# Unified Biomolecular Trajectory Generation via Pretrained Variational Bridge

**Ziyang Yu**[1,2] **Wenbing Huang**[3,4,*] **Yang Liu**[1,2,*]

[1]Department of Computer Science and Technology, Tsinghua University
[2]Institute for AI Industry Research (AIR), Tsinghua University
[3]Gaoling School of Artificial intelligence, Renmin University of China
[4]Beijing Key Laboratory of Research on Large Models and Intelligent Governance
`yu-zy24@mails.tsinghua.edu.cn, hwenbing@ruc.edu.cn, liuyang2011@tsinghua.edu.cn`

## Abstract

Molecular Dynamics (MD) simulations provide a fundamental tool for characterizing molecular behavior at full atomic resolution, but their applicability is severely constrained by the computational cost. To address this, a surge of deep generative models has recently emerged to learn dynamics at coarsened timesteps for efficient trajectory generation, yet they either generalize poorly across systems or, due to limited molecular diversity of trajectory data, fail to fully exploit structural information to improve generative fidelity. Here, we present the Pretrained Variational Bridge (PVB) in an encoder-decoder fashion, which maps the initial structure into a noised latent space and transports it toward stage-specific targets through augmented bridge matching. This unifies training on both single-structure and paired trajectory data, enabling consistent use of cross-domain structural knowledge across training stages. Moreover, for protein-ligand complexes, we further introduce a reinforcement learning-based optimization via adjoint matching that speeds progression toward the holo state, which supports efficient post-optimization of docking poses. Experiments on proteins and protein-ligand complexes demonstrate that PVB faithfully reproduces thermodynamic and kinetic observables from MD while delivering stable and efficient generative dynamics.

## 1 Introduction

Molecular Dynamics (MD) simulations are indispensable ingredients in chemistry, materials, drug discovery, and other fields (Van Gunsteren & Berendsen, 1990; Lindorff-Larsen et al., 2011; Hollingsworth & Dror, 2018; Lau et al., 2018). Given the molecular system of interest within a particular chemical environment, the essence of MD is to obtain the conformational ensemble that follows laws of physics through *in silico* simulation, which provides accurate estimates of kinetic (*e.g.*, dissociation rate constant) and thermodynamic (*e.g.*, free energy) observables (Shaw et al., 2010; Deng & Roux, 2009; Wang et al., 2023; Chipot, 2023).

Under certain conditions, the equilibrium measure of molecular ensembles follows the Boltzmann distribution (Boltzmann, 1877). To obtain such equilibrium samples in practice, classical MD simulations widely rely on Langevin dynamics (Langevin, 1908), which models the temporal evolution of the system through a Stochastic Differential Equation (SDE) and induces a Markov-chain sampling process whose stationary distribution corresponds to the Boltzmann distribution (Paquet & Viktor, 2015). Although these methods have been firmly established in practice, the stability of numerical integration requires exceedingly small time steps $\Delta t$ (around 1 femtosecond), making the computational cost prohibitively high for long-timescale or high-throughput simulations (Schlick, 2001).

Recently, the rise of deep learning methods has opened up new possibilities for boosting MD simulations and ensemble generation. In contrast to directly fitting the Boltzmann distribution that ignores temporal correlations, we focus on *time-coarsened dynamics*, which aims to fit the

---

*Corresponding authors: Wenbing Huang, Yang Liu.

conditional distribution $\mu(x_{t+\tau} \mid x_t)$, defined as the transition density of the underlying MD process that maps a current state $x_t$ to its distribution after a lag $\tau \gg t$, thereby enabling time-dependent trajectory generation on a coarse-grained temporal scale (Schreiner et al., 2023; Klein et al., 2023; Jing et al., 2024b; Yu et al., 2025). This paradigm achieves substantial efficiency gains while preserving both the kinetic and thermodynamic properties of the system. However, several challenges remain in the literature. First, the stability of trajectory generation over long timescales cannot yet be reliably ensured. In addition, most existing methods are confined to one specific molecular domain (*e.g.*, proteins) and therefore fail to model cross-domain systems. In particular, UniSim (Yu et al., 2025) employs 3D molecular pretraining to obtain a unified atomic representation model, demonstrating strong generalization for cross-domain MD simulations. However, the inconsistency between pretraining on single structures $x$ and finetuning on trajectory pairs $(x_t, x_{t+\tau})$ could result in suboptimal transfer and hinder the effective use of pretrained knowledge. A further limitation of most existing methods is their focus on single-molecule simulations, while multi-molecular systems such as protein-ligand complexes have been much less explored.

To address the aforementioned issues, we propose the Pretrained Variational Bridge (PVB), a novel generative model that leverages an encoder-decoder architecture to provide a unified framework for both pretraining and finetuning. Specifically, we first enhance the cross-domain generalization by pretraining on a large variety of high-resolution single-structure data $x$, and then finetune on the paired data $(x_t, x_{t+\tau})$ drawn from MD trajectories to fit the conditional density $\mu(x_{t+\tau} \mid x_t)$ with a predefined time step $\tau$. The key challenge is to elegantly resolve the objective mismatch between unconditional generation during pretraining and conditional generation during finetuning. To this end, we introduce two random variables, $\mathbf{X}_0$ and $\mathbf{Y}_1$, and define a conditional probability measure $q(\mathrm{d}\mathbf{Y}_1 \mid \mathbf{X}_0)$ to be learned across training stages. For single-structure data $x$, this measure is degenerate, assigning all probability mass to the point $\mathbf{Y}_1 = x$ given $\mathbf{X}_0 = x$. For finetuning on paired MD data $(x_t, x_{t+\tau})$, $q(\mathrm{d}\mathbf{Y}_1 \mid \mathbf{X}_0 = x_t)$ is defined such that its con-

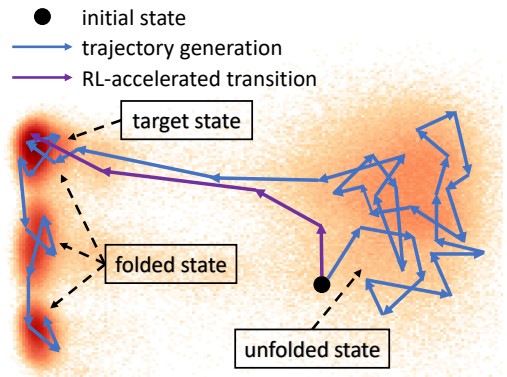

Figure 1: Two variants of PVB. Blue arrows show coarse-grained sampling that traverses unfolded and folded states to reproduce the Boltzmann distribution, while purple arrows denote accelerated transition that drives rapid access from the initial state to the target folded state via reinforcement learning.

ditional density matches the target transition density $\mu(x_{t+\tau} \mid x_t)$ of the underlying MD process. Since the measure collapses to a point mass in the single-structure pretraining scenario, we further introduce a latent random variable $\mathbf{Y}_0$ to decompose the measure and prevent the model from degenerating into this trivial case:

$$q(\mathbf{Y}_1 \in A \mid \mathbf{X}_0) \coloneqq \int q_d(\mathbf{Y}_1 \in A \mid \mathbf{Y}_0) q_e(\mathrm{d}\mathbf{Y}_0 \mid \mathbf{X}_0), \text{ for any measurable set } A. \quad (1)$$

Here $q_e$ and $q_d$ denote the transition kernels of the Markov chain $\mathbf{X}_0 \to \mathbf{Y}_0 \to \mathbf{Y}_1$, which are respectively parametrized by an encoder and a separate generative decoder using *augmented bridge matching* (De Bortoli et al., 2023). The process of mapping the initial state to the noised latent space and back-projecting it to target states enables the model to acquire cross-domain structural knowledge in pretraining, which, due to objective consistency, greatly facilitates subsequent finetuning.

Furthermore, to better capture protein-ligand dynamics, we incorporate a finetuning procedure into the generative framework based on Reinforcement Learning (RL). By following *adjoint matching* (Domingo-Enrich et al., 2024), we provide a memory-efficient finetuning scheme with explicit reward functions. The RL optimization is designed to modulate the generative distribution, guiding it toward the holo state and thereby enabling rapid evolution from the apo state within short generated trajectories. Figure 1 illustrates the two generative modes of PVB described above.

Overall, our contributions in this work are summarized as below:

- We present PVB, a novel generative model that integrates the encoder-decoder architecture with augmented bridge matching, offering a unified training framework for both single-structure and paired trajectory data to leverage the rich knowledge from pretraining.

- We propose an RL-based finetuning procedure, applying the stochastic optimal control paradigm to augmented bridge matching. For the protein-ligand flexible docking task, PVB learns to bypass inefficient local exploration and rapidly evolve toward the holo state within short simulations, making it an efficient tool for post-optimization of docking poses.

- For both protein monomers and protein-ligand complexes, PVB achieves comparable results to MD in terms of thermodynamic and kinetic metrics, while delivering substantial improvements over the baseline in generation stability.

## 2 BACKGROUND

**Ensemble Generation** Deep learning techniques have introduced methodological innovations to ensemble generation. One promising approach is to replace empirical force fields with accurate all-atom neural network potentials to accelerate simulations of large biomolecular systems, with representative work exemplified by AI$^2$BMD (Wang et al., 2024b). This approach, however, does not overcome the inherent limitations of sequential trajectory generation with small time steps. To move beyond the framework of Markov chain sampling, a surge of methods leverage generative models to fit the empirical distribution from MD trajectories (Jing et al., 2024a; Wang et al., 2024a; Lewis et al., 2025), while *Boltzmann generators* further reproduce the Boltzmann distribution from the surrogate model by importance sampling (Noé et al., 2019; Klein & Noé, 2024; Tan et al., 2025). Such methods enable highly efficient parallel generation of i.i.d. samples, yet the absence of temporal dependencies blocks further estimation of kinetic observables.

**Trajectory Generation** The paradigm of trajectory generation, also referred to in the literature as *time-coarsened dynamics*, seeks to learn the conditional distribution $\mu(x_{t+\tau} \mid x_t)$ at a coarse time step $\tau \gg \Delta t$, enabling the recovery of observables of interest from relatively short generated trajectories (Schreiner et al., 2023; Klein et al., 2023; Jing et al., 2024b; Yu et al., 2025). Notably, UniSim (Yu et al., 2025) first extends the task to cross-domain biomolecules, showcasing strong generalizability with the pretrained atomic representation model and force-guided finetuning techniques. However, due to the inconsistency in training objectives, the pretraining stage produces only a unified atomic representation model, failing to explicitly capture cross-domain structural information. In contrast, our approach integrates the generative model into pretraining via a noised latent space, which provides a unified framework for training on both single-structure data and paired MD trajectories, enabling the exploitation of rich structural knowledge across training stages.

## 3 METHOD

We now discuss the methodology and rationale of PVB, with its overall workflow illustrated in Figure 2. We first present the problem formulation with necessary notations in Section 3.1. Next, in Section 3.2, we discuss the overall generative framework of our pretrained variational bridge, which provides a unified training interface for single-structure data and paired trajectory data. Moreover, in Section 3.3, we develop a memory-efficient RL-based finetuning framework grounded in stochastic optimal control, which can be applied to accelerate exploration for the holo state of protein-ligand complexes. Details of the model architecture are shown in Section C.

### 3.1 TASK FORMULATION

In this work, the conformation of one molecular system is represented in a simplified triplet form $(\boldsymbol{z}, \boldsymbol{C}, x)$, where $\boldsymbol{z} \in \mathbb{N}^N$ denotes the atomic numbers of $N$ heavy atoms, $\boldsymbol{C} \in \mathbb{N}^{2 \times C}$ represents $C$ covalent bonds among heavy atoms in the molecular topology, and $x \in \mathbb{R}^{N \times 3}$ corresponds to the Euclidean coordinates of all heavy atoms[1]. We consider two types of molecular systems: (I) systems represented by one or a few independent conformations, denoted as $(\boldsymbol{z}, \boldsymbol{C}, \{x\})$; and (II) systems represented by time-dependent MD trajectories, denoted as $(\boldsymbol{z}, \boldsymbol{C}, \{x_{\Delta t}, x_{2\Delta t}, \cdots\})$, where $\Delta t$ is

---

[1]Lowercase is used for coordinates to distinguish them from random variable notations.

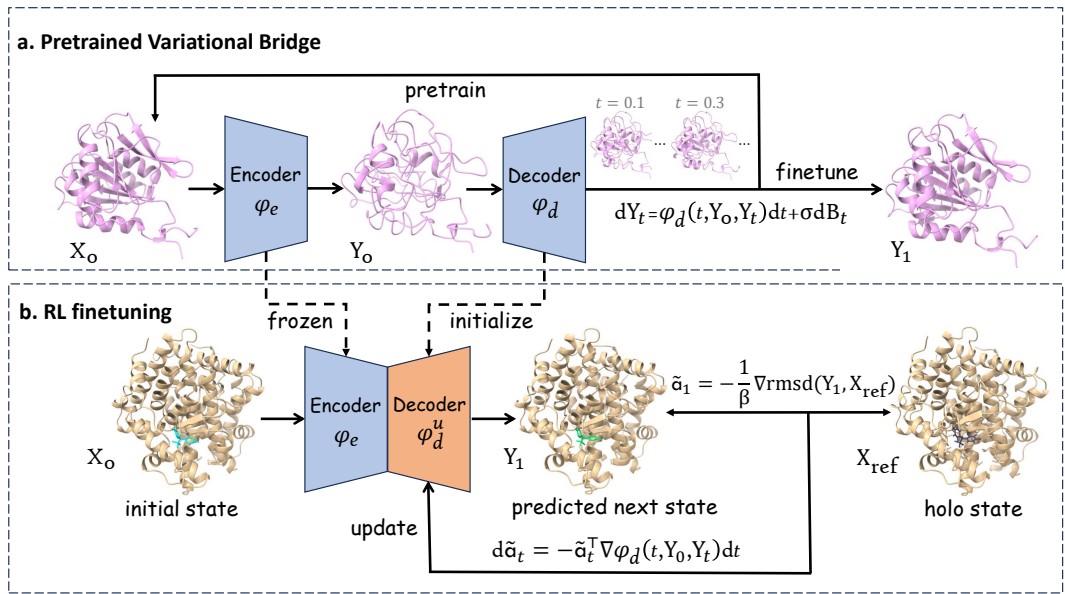

Figure 2: Schematic of the overall PVB workflow. **a.** The unified framework for pretraining on single-structure data and finetuning on paired trajectory data. The encoder $\varphi_e$ maps the initial state $\mathbf{X}_0$ to the latent variable $\mathbf{Y}_0$, which the decoder $\varphi_d$ then propagates to the stage-specific target $\mathbf{Y}_1$ via Equation (6). **b.** RL finetuning with frozen $\varphi_e$ and $\varphi_d^u$ initialized from the finetuned $\varphi_d$, aiming to accelerate exploration of protein-ligand holo states. Given the predicted next state $\mathbf{Y}_1$ generated by $\varphi_e$ and $\varphi_d^u$, the *lean adjoint state* $\tilde{a}_1$ is first computed from the reward function (*i.e.*, the root mean square error to the holo state $\mathbf{X}_{\text{ref}}$), and then propagated backward to $\tilde{a}_t$ by solving Equation (12) with diffusion time $t$, which is subsequently used to update $\varphi_d^u$ according to Equation (11).

the integration time step, which may vary across datasets. Our tasks are: (i) to pretrain on type-I high-resolution structural data to acquire cross-domain full-atom generative capability, and (ii) to finetune on type-II MD trajectory data to learn dynamics for trajectory generation, with appropriate framework and training objectives to maximally leverage the pretrained structural knowledge.

## 3.2 PRETRAINED VARIATIONAL BRIDGE

First and foremost, our primary objective is to endow the model with the cross-domain generalizability of biomolecular trajectory generation. Breakthroughs are difficult to achieve by training solely on MD data due to two key limitations: the high cost of generating accurate MD trajectories and the poor accuracy/adaptability of data from empirical force fields across diverse molecular domains. Inspired by UniSim (Yu et al., 2025), we aim to first pretrain on extensive high-resolution biomolecular structures to fully exploit structural information within a generative framework, and then seamlessly transfer this knowledge to trajectory generation through carefully designed finetuning objectives.

Formally, we model the generation process in both pretraining and finetuning as a Markov chain $\mathbf{X}_0 \to \mathbf{Y}_0 \to \mathbf{Y}_1$. For single-structure data (type-I systems) used in pretraining, we set the initial and target states as $(\mathbf{X}_0, \mathbf{Y}_1) = (x, x)$. For paired trajectory data (type-II systems) used in finetuning, we set $(\mathbf{X}_0, \mathbf{Y}_1) = (x_t, x_{t+\tau})$. The latent variable $\mathbf{Y}_0$ is introduced to prevent the conditional law $q(\mathrm{d}\mathbf{Y}_1 \mid \mathbf{X}_0)$ from collapsing to a Dirac measure for type-I systems.

According to Equation (1), the target conditional law $q(\mathrm{d}\mathbf{Y}_1 \mid \mathbf{X}_0)$ is realized as the composition of two Markov kernels: an encoder kernel $q_e(\mathrm{d}\mathbf{Y}_0 \mid \mathbf{X}_0)$ and a decoder kernel $q_d(\mathrm{d}\mathbf{Y}_1 \mid \mathbf{Y}_0)$. We employ an encoder-decoder architecture to parametrize these two distributions separately, as detailed below.

**Variational Encoder** Given the initial state $x_0$, the conditional probability measure of the latent variable $\mathbf{Y}_0$ is manually specified by $q_e(\mathrm{d}\mathbf{Y}_0 \mid \mathbf{X}_0 = x_0) \coloneqq \mathcal{N}(x_0, \sigma_e^2 \boldsymbol{I})$, where $\sigma_e$ is the hyperparameter of deviation. Choosing an appropriate prior distribution is crucial for performance, as it

should preserve sufficient structural information through encoding, and also avoid collapse of the decoding process into trivial cases (*e.g.*, a Dirac measure). To this end, we use $\sigma_e = \sqrt{0.5}$Å in our experiments, which is considerably larger than those typically adopted in 3D molecular pretraining methods (Zaidi et al., 2023; Jiao et al., 2024; Yu et al., 2025).

Next, a neural network encoder $\varphi_e$ is leveraged to minimize the distance between the learned distribution $p_e(\mathrm{d}\mathbf{Y}_0 \mid \mathbf{X}_0)$ and the prior $q_e(\mathrm{d}\mathbf{Y}_0 \mid \mathbf{X}_0)$ under the Kullback-Leibler (KL) divergence measure, with its architecture discussed in Section C. By applying the reparameterization trick (Kingma & Welling, 2013), the encoder will yield the logarithm of the predicted variance $\log \mathbf{V} = \varphi_e(\mathbf{X}_0) \in \mathbb{R}^N$ [2]. Then the KL divergence loss is given by:

$$\mathcal{L}_{\mathrm{KL}} := \mathbb{E}_{\mathbf{X}_0}[D_{\mathrm{KL}}(p_e(\cdot \mid \mathbf{X}_0)\|q_e(\cdot \mid \mathbf{X}_0))] = -\frac{1}{2}\mathbb{E}[1 + \log \mathbf{V} - 2\log \sigma_e - \frac{\mathbf{V}}{\sigma_e^2}]. \tag{2}$$

**Augmented Bridge Decoder**  Given the latent variable $\mathbf{Y}_0 \sim p_e(\cdot \mid \mathbf{X}_0)$, we utilize a separate decoder $\varphi_d$ to fit the transition kernel $q_d(\mathrm{d}\mathbf{Y}_1 \mid \mathbf{Y}_0)$. Since the decoder stage is critical to model performance, we adopt the augmented bridge matching framework (De Bortoli et al., 2023), which ensures preservation of the coupling between the latent variable $\mathbf{Y}_0$ and the target state $\mathbf{Y}_1$, denoted by $\Pi_{0,1}$.

Specifically, we consider the path measure $\mathbb{Q} : \mathbb{C}([0,1], \mathbb{R}^{N \times 3}) \to \mathbb{R}_+$ associated with the Brownian motion $(\sigma \mathbf{B}_t)_{t \in [0,1]}$, with $\sigma \in \mathbb{R}_+$ a predefined noise schedule. Next, we define the twisted path measure $\mathbb{P} = \Pi_{0,1}\mathbb{Q}_{|0,1}$, where the trajectory $\mathbf{Y}_{0:1} \sim \mathbb{P}$ can be generated by first sampling $(\mathbf{Y}_0, \mathbf{Y}_1)$ from the coupling $\Pi_{0,1}$, then sampling from the Brownian bridge $\mathbb{Q}_{|0,1}(\cdot \mid \mathbf{Y}_0, \mathbf{Y}_1)$ pinned down on the two endpoints. As a well-studied case in the literature, the temporal marginal of $\mathbb{P}$ at diffusion time $t$ and the associated forward SDE are given by:

$$\mathbf{Y}_t = t\mathbf{Y}_1 + (1-t)\mathbf{Y}_0 + \sigma\sqrt{t(1-t)}\mathbf{B}_t, \ (\mathbf{Y}_0, \mathbf{Y}_1) \sim \Pi_{0,1}, \tag{3}$$

$$\mathrm{d}\mathbf{Y}_t = \frac{\mathbf{Y}_1 - \mathbf{Y}_t}{1-t}\mathrm{d}t + \sigma\,\mathrm{d}\mathbf{B}_t, \ (\mathbf{Y}_0, \mathbf{Y}_1) \sim \Pi_{0,1}. \tag{4}$$

De Bortoli et al. (2023) show that one can preserve the coupling $\Pi_{0,1}$ by first training a parameterized vector field $\varphi_d$ conditioned on $t$, $\mathbf{Y}_0$, and $\mathbf{Y}_t$ to minimize the quadratic loss:

$$\mathcal{L}_{\mathrm{ABM}} = \mathbb{E}_{t \sim \mathcal{U}(0,1),(\mathbf{Y}_0,\mathbf{Y}_1)\sim\Pi_{0,1}}\left[\mathbb{E}_{\mathbf{Y}_t \sim \mathbb{Q}_{t|0,1}}\left[\left\|\varphi_d(t, \mathbf{Y}_0, \mathbf{Y}_t) - \frac{\mathbf{Y}_1 - \mathbf{Y}_t}{1-t}\right\|^2\right]\right], \tag{5}$$

where $\mathcal{U}(0,1)$ denotes the uniform distribution over $[0,1]$. Afterwards, simulating the resulting non-Markovian SDE ensures that the coupling $\Pi_{0,1}$ is preserved:

$$\mathrm{d}\mathbf{Y}_t = \varphi_d^*(t, \mathbf{Y}_0, \mathbf{Y}_t)\,\mathrm{d}t + \sigma\,\mathrm{d}\mathbf{B}_t, \ \mathbf{Y}_0 \sim \Pi_0, \tag{6}$$

where $\Pi_0$ is the marginal probability measure of $\mathbf{Y}_0$, and $\varphi_d^*$ is the minimizer of Equation (5). Finally, Proposition 1 demonstrates that our proposed encoder-decoder architecture provides an unbiased estimate of the target conditional law $q(\mathrm{d}\mathbf{Y}_1 \mid \mathbf{X}_0)$ (see Section B):

**Proposition 1.** *Denote by $p_e^*$ the minimizer of Equation* (2)*, $\mathbb{P}^*$ the path measure associated with Equation* (6)*, and $p_d^*$ the probability density function of the conditional measure $\mathbb{P}_{1|0}^*$. Then the following equality holds:*

$$q(\mathbf{Y}_1 \in A \mid \mathbf{X}_0) = \int p_d^*(\mathbf{Y}_1 \in A \mid \mathbf{Y}_0)p_e^*(\mathrm{d}\mathbf{Y}_0 \mid \mathbf{X}_0), \textit{ for any measurable set } A. \tag{7}$$

Overall, the objective $\mathcal{L} = w_{\mathrm{KL}} \cdot \mathcal{L}_{\mathrm{KL}} + w_{\mathrm{ABM}} \cdot \mathcal{L}_{\mathrm{ABM}}$ is employed for both pretraining and finetuning, where $w_{\mathrm{KL}}$ and $w_{\mathrm{ABM}}$ are hyperparameters that control the relative weights.

## 3.3  MEMORY-EFFICIENT STOCHASTIC OPTIMAL CONTROL

In this section, we further explore the application of the generated trajectories to protein-ligand flexible docking. Such tasks focus on locating the global minimum-energy conformation, rather

---

[2]For convenience, we omit $\mathbf{z}$ and $\mathbf{C}$ from the neural network parameters, and the same applies hereafter.

than reconstructing the full Boltzmann distribution. Notably, the protein-ligand interaction can occur on the millisecond timescale or longer, making its simulation computationally prohibitive even for models with coarsened timesteps. To address this limitation, we integrate an RL-based finetuning procedure with an explicit reward function $r(x)$, designed to adjust the generative distribution and thereby accelerate the evolution of trajectories toward the holo state.

Formally, we first consider the following SDE with a control vector field $u$ to be optimized:

$$\mathrm{d}\mathbf{Y}_t = (\varphi_d^*(t, \mathbf{Y}_0, \mathbf{Y}_t) + \sigma u(t, \mathbf{Y}_0, \mathbf{Y}_t))\,\mathrm{d}t + \sigma\,\mathrm{d}\mathbf{B}_t,\ \mathbf{Y}_0 \sim \Pi_0. \tag{8}$$

Denote by $\mathbb{P}^u$ the path measure associated with Equation (8). We aim to solve the optimization problem with a KL-regularized objective:

$$\max_u \mathbb{E}_{\mathbf{Y}_0 \sim \Pi_0}[\mathbb{E}_{\mathbf{Y}_{0:1} \sim \mathbb{P}^u_{|0}(\cdot|\mathbf{Y}_0)}[r(\mathbf{Y}_1) - \beta \cdot D_{\mathrm{KL}}(\mathbb{P}^u_{|0}(\mathbf{Y}_{0:1} \mid \mathbf{Y}_0)\|\mathbb{P}^*_{|0}(\mathbf{Y}_{0:1} \mid \mathbf{Y}_0))]], \tag{9}$$

where $\beta$ controls the regularization strength. Based on the Girsanov theorem, the KL divergence term can be expressed by the control vector field $u$ (see Section B):

$$\max_u \mathbb{E}_{\mathbf{Y}_0 \sim \Pi_0}\left[\mathbb{E}_{\mathbf{Y}_{0:1} \sim \mathbb{P}^u_{|0}(\cdot|\mathbf{Y}_0)}\left[r(\mathbf{Y}_1) - \frac{\beta}{2} \cdot \int_0^1 \|u(t, \mathbf{Y}_0, \mathbf{Y}_t)\|^2\,\mathrm{d}t\right]\right]. \tag{10}$$

Although this training objective is already applicable for model optimization, it requires gradient accumulation during simulation along the SDE, which leads to prohibitive memory overhead. Hopefully, by first considering the inner expectation in Equation (10) given that $\mathbf{Y}_0$ takes the value $y_0$, the dependence on $\mathbf{Y}_0$ in $u(t, \mathbf{Y}_0, \mathbf{Y}_t)$ can be treated as fixed, casting the optimization problem into the standard Stochastic Optimal Control (SOC) paradigm (Bellman, 1966). Based on the conclusions of adjoint matching (Domingo-Enrich et al., 2024), we state the following proposition (see Section B):

**Proposition 2.** *Assume that the function class of $u$ is rich enough so that the value of $u(t, y_0, \cdot)$ can be chosen independently for each $y_0$, then the unique minimizer $u^*$ of the following objective is the optimal control of Equation* (10)*:*

$$\mathcal{L}_{\mathrm{adj}} = \mathbb{E}_{\mathbf{Y}_0 \sim \Pi_0}\left[\mathbb{E}_{t \sim \mathcal{U}(0,1), \mathbf{Y}_{0:1} \sim \mathbb{P}^{\bar{u}}_{|0}(\cdot|\mathbf{Y}_0)}\left[\|u(t, \mathbf{Y}_0, \mathbf{Y}_t) + \sigma\tilde{a}(t, \mathbf{Y}_{0:1})\|^2\right]\right],\ \bar{u} = \mathrm{sg}(u), \tag{11}$$

$$\text{where } \frac{\mathrm{d}}{\mathrm{d}s}\tilde{a}(s, \mathbf{Y}_{0:1}) = -\tilde{a}(s, \mathbf{Y}_{0:1})^\top \nabla_{\mathbf{Y}_s}\varphi_d^*(s, \mathbf{Y}_0, \mathbf{Y}_s),\ \tilde{a}(1, \mathbf{Y}_{0:1}) = -\frac{1}{\beta}\nabla_{\mathbf{Y}_1}r(\mathbf{Y}_1). \tag{12}$$

Here $\mathrm{sg}(\cdot)$ denotes the stop-gradient operator, and the random variable $\tilde{a}$ is known in the literature as the *lean adjoint state*. Denote by $\varphi_d^u$ the neural network optimized during RL finetuning, with parameters initialized from $\varphi_d^*$. In light of Equation (6) and Equation (8), the control vector field can be reparameterized by $u = \frac{1}{\sigma}(\varphi_d^u - \varphi_d^*)$. This formulation avoids the need to introduce any additional networks. For clarity, we provide the pseudocode of RL finetuning in Algorithm 1.

## 4 EXPERIMENT

In this section, we provide details on the application of PVB to trajectory generation and related tasks.

**Dataset** For pretraining, we use a variety of datasets covering different molecular domains: PCQM4Mv2 (Hu et al., 2021) and ANI-1x (Smith et al., 2020) for small organic molecules, PDB (Berman et al., 2000) for protein monomers, and PDBBind2020 (Wang et al., 2004; 2005; Liu et al., 2015) for protein-ligand complexes. After pretraining, the model is finetuned on domain-specific datasets: ATLAS (Vander Meersche et al., 2024) and mdCATH (Mirarchi et al., 2024) for protein monomers, and MISATO (Siebenmorgen et al., 2024) for protein-ligand complexes, enabling trajectory generation in these domains. Additionally, we augment the PDBBind2020 protein-ligand complex data with short MD simulations using `OpenMM` (Eastman et al., 2023), which are further employed for RL finetuning to explore protein-ligand holo states. Comprehensive details of the datasets and preprocessing procedures are provided in Section D.

**Baselines** The following deep learning-based models tailored for trajectory generation are selected as baselines: (I) **ITO** (Schreiner et al., 2023), a conditional diffusion model that fits the transition

kernel of molecular dynamics with varying temporal scales. (II) **MDGEN** (Jing et al., 2024b), a generative model based on torus and local frame representations of proteins, which can be applied to diverse dynamics-related tasks. (III) **UniSim** (Yu et al., 2025), a pretrained generative model with force-guided finetuning modules, which exhibits cross-domain transferability. (IV) **AlphaFlow** (Jing et al., 2024a), a modification of AlphaFold (Jumper et al., 2021) with the flow-matching objective for finetuning, enabling the generation of independent and identically distributed (i.i.d.) protein conformational ensembles. For a fair comparison, we follow the standard training procedures of each baseline: ITO and MDGEN are trained from scratch, UniSim is initialized from its officially released pretrained representation model, and AlphaFlow is initialized from the publicly released AlphaFold pretrained weights, all subsequently finetuned on our dataset.

## 4.1 TRAJECTORY GENERATION

**Proteins on ATLAS** We now investigate the capability of the model to generate trajectories on the protein domain. Building on the pretrained PVB parameters, we finetune on the ATLAS dataset to capture dynamic patterns, with 790/14 systems for training and testing, respectively. Specifically, for each 100 ns trajectory in the training set, we randomly sample 500 data pairs $(x_t, x_{t+\tau})$ from the first 80 ns for training and 100 data pairs from the last 20 ns for validation, with the coarsened time step $\tau = 100$ ps. Given the finetuned model, we generate trajectories for the 14 test protein systems by simulating Equation (6) via numerical discretization, with a chain length of 1,000.

To comprehensively evaluate the generated trajectory, we apply the following metrics: (I) the Jensen-Shannon Divergence (JSD) calculated on projected feature spaces, including the radius of gyration (Rg), the torsion angles $\phi$ and $\psi$ of protein backbones (Torus), the slowest two Time-lagged Independent Components (TIC0 and TIC1) (Pérez-Hernández et al., 2013), and the state occupancy estimated Markov state models (MSM). (II) The proportion of conformations without bond break or clash between C$\alpha$ atoms, denoted as VAL-CA. (III) The root mean square error of contact maps between generated trajectories and reference MD trajectories, denoted as RMSE-CONTACT. (IV) The decorrelation fraction with respect to TIC0 observed in the generated trajectories, termed as Decorr-TIC0. The reference statistics for each metric are derived from three replicate MD trajectories of each test system in ATLAS. Further details on how the metrics are computed can be found in Section F.

To facilitate comparison, the first 10 ns of one selected MD trajectory was included in the analysis for each test system. As presented in Table 1, PVB achieves performance comparable to the baselines across most metrics, with clear superiority in the validity of generated ensembles. Moreover, compared to 10 ns MD, PVB maintains equivalent physical realism and achieves superior distributional agreement in TIC and MSM feature spaces. This is further evidenced by a higher decorrelation ratio, confirming its ability to resolve slow dynamical modes inaccessible to short MD simulations.

Table 1: Results on the ATLAS test set, with each metric reported as mean/std across 14 test protein monomers. The best and second-best results for each metric are shown in **bold** and underlined, respectively. * denotes models that generate i.i.d. protein samples instead of trajectories.

| MODELS | JSD ($\downarrow$) | | | | VAL-CA ($\uparrow$) | RMSE-CONTACT ($\downarrow$) | Decorr-TIC0 ($\uparrow$) |
| --- | --- | --- | --- | --- | --- | --- | --- |
| | Rg | Torus | TIC | MSM | | | |
| MD (10 ns) | 0.379/0.115 | 0.148/0.016 | 0.684/0.072 | 0.596/0.179 | 0.926/0.093 | 0.040/0.010 | 0.000 |
| AlphaFlow* | **0.385**/0.143 | 0.439/0.045 | 0.570/0.110 | 0.394/0.116 | 0.485/0.386 | **0.107**/0.068 | - |
| ITO | 0.792/0.022 | 0.560/0.018 | 0.400/0.081 | 0.469/0.145 | 0.001/0.000 | 0.948/0.016 | 0.714 |
| MDGEN | 0.493/0.115 | **0.321**/0.031 | 0.400/0.085 | 0.463/0.150 | 0.098/0.217 | 0.158/0.020 | 0.857 |
| UniSim | 0.538/0.135 | 0.344/0.012 | 0.372/0.078 | 0.344/0.097 | 0.106/0.050 | 0.129/0.019 | 0.786 |
| PVB (*ours*) | 0.457/0.087 | 0.336/0.030 | **0.371**/0.087 | **0.333**/0.105 | **0.975**/0.023 | 0.143/0.031 | **0.929** |

**Proteins on mdCATH** We further evaluate the model on the mdCATH dataset, which encompasses a wider range of protein domains. Following Janson et al. (2025), we use a 5,049/40 split for training and validation, with 20 domains selected from the original test set for evaluation. Training data pairs $(x_t, x_{t+\tau})$ are sampled every 5 ns, with the coarsened time step $\tau = 1$ ns. After finetuning, we generate 500-step trajectories for each of the 20 test systems, matching the standard trajectory length in mdCATH. For evaluation, four of the five replicate MD trajectories per system are used to construct reference statistics, while the fifth serves as the MD oracle for direct comparison with PVB.

The evaluation results on mdCATH are shown in Table 2. Similarly, PVB achieves MD-level physical plausibility in the generated conformations and shows good agreement with the slow modes (TIC)

and metastable states (MSM) observed in the MD reference ensembles. Moreover, PVB outperforms the baseline on most metrics and, compared to the ATLAS results, exhibits a greater disparity in distributional similarity for projected features, thereby demonstrating enhanced generalization on a broader-scope dataset.

Table 2: Results on the test set of mdCATH. Values of each metric are shown in mean/std of all 20 test protein domains. The best result for each metric is shown in **bold**.

| MODELS | JSD ($\downarrow$) | | | | VAL-CA ($\uparrow$) | RMSE-CONTACT ($\downarrow$) | Decorr-TIC0 ($\uparrow$) |
|---|---|---|---|---|---|---|---|
| | Rg | Torus | TIC | MSM | | | |
| MD Oracle | 0.264/0.085 | 0.182/0.037 | 0.316/0.071 | 0.282/0.069 | 0.942/0.154 | 0.067/0.039 | 0.70 |
| UniSim | 0.562/0.160 | 0.357/0.050 | 0.454/0.100 | 0.402/0.114 | 0.522/0.371 | **0.116**/0.041 | 0.20 |
| PVB (*ours*) | **0.437**/0.142 | **0.347**/0.035 | **0.376**/0.050 | **0.362**/0.124 | **0.963**/0.049 | 0.177/0.030 | **0.90** |

Figure 3 presents visualizations for four representative test systems, two from ATLAS and two from mdCATH. It demonstrates that the generated trajectories faithfully recover the underlying free energy landscapes, while the metastable state occupancies estimated by the MSM show strong agreement with the reference values. Together, these findings highlight the reliability of PVB in reliably capturing key dynamical properties during the long-time evolution of large molecular systems.

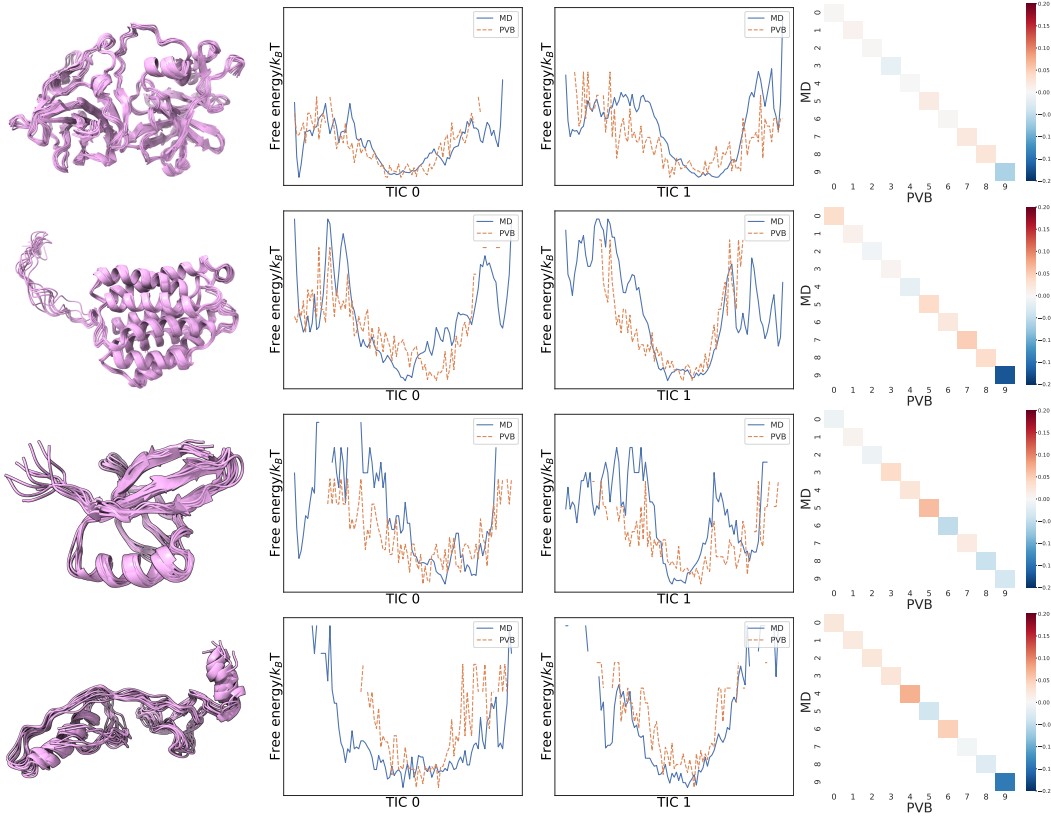

Figure 3: Illustration of generated trajectories for PDB 2bjq (row 1), PDB 7rm7 (row 2), CATH domain 3er0A02 (row 3), and CATH domain 1pyaA00 (row 4). **Left**: Representative structures from the first 10 frames of the generated trajectories. **Middle**: Free energy surfaces projected onto TIC0 and TIC1, respectively. **Right**: Probability differences between PVB and MD across the 10 metastable states estimated by MSM.

**Protein-Ligand Complexes on MISATO** Next, we investigate the application to protein-ligand complexes using the MISATO dataset, with 13,765/1,595/20 systems for training, validation and testing, respectively. For the 8 ns trajectory comprising 100 snapshots, all frames are utilized for

finetuning, with a lag time of $\tau = 80$ ps. For consistency with the MD data, we generate trajectories of length 100 for evaluation.

Considering the characteristics of protein-ligand complexes, we adopt more specialized evaluation metrics: (I) the Wasserstein-1 distance (Villani et al., 2008) of the ligand RMSD relative to the MD initial state after superposition onto the pocket backbone (EMD-ligand), as well as the distances between the center of mass of the pocket and the ligand (EMD-CoM). All distances are reported in nanometers. (II) The root mean square error of the contact occupancy between heavy atoms of the ligand and the pocket, also denoted as RMSE-CONTACT. Further details can be found in Section F.

The results in Table 3 show that, PVB achieves the closest match in ligand poses and pocket-ligand center-of-mass distances compared to MD trajectories, while also reaching comparable performance in heavy-atom contacts. Notably, the EMD corresponds to errors on the order of atomic resolution, demonstrating the accurate characterization of protein-ligand complex interactions.

Table 3: Results on the test set of MISATO. Values of each metric are shown in mean/std of all 20 test protein-ligand complexes. The best result for each metric is shown in **bold**.

| MODELS | EMD-ligand ($\downarrow$) | EMD-CoM ($\downarrow$) | RMSE-CONTACT ($\downarrow$) |
|---|---|---|---|
| ITO | 0.494/0.240 | 0.479/0.258 | 0.987/0.001 |
| UniSim | 0.196/0.137 | 0.128/0.111 | **0.049**/0.009 |
| PVB (*ours*) | **0.133**/0.072 | **0.089**/0.078 | 0.055/0.008 |

Meanwhile, Figure 4 presents visualizations of ligand RMSD and pocket-ligand CoM distance for two representative cases. It can be observed that the trajectories generated by PVB closely match MD results in both distribution and temporal evolution, showing that the model accurately captures the dynamical behaviors of the molecular system.

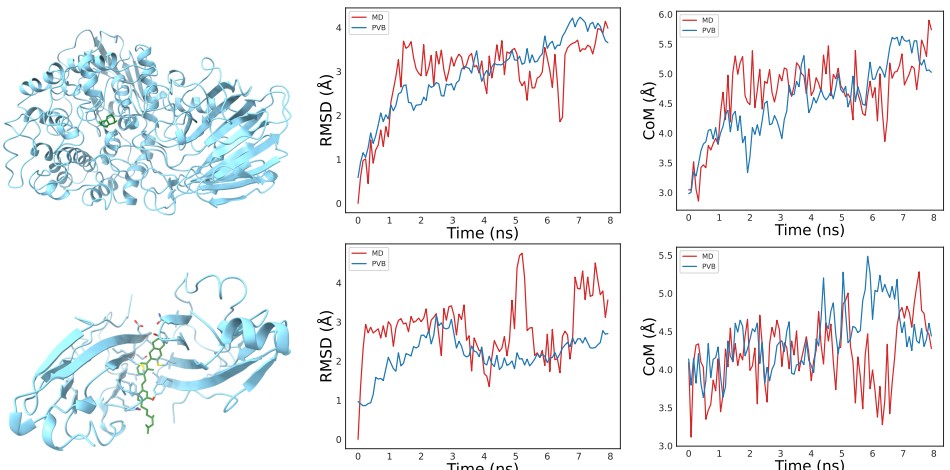

Figure 4: Illustration of ligand RMSD and CoM along the trajectories for PDB 2ww0 (top row) and PDB 5n2f (bottom row). **Left**: Visualization of the complex, with the receptor shown in blue and the ligand in green. **Middle**: Ligand RMSD over time. **Right**: Pocket-ligand CoM distance over time.

## 4.2 EXPLORATION FOR PROTEIN-LIGAND HOLO STATE

We further evaluate the model-generated trajectories for exploring the holo state of protein-ligand complexes. Generating trajectories directly from the apo to the holo state is computationally prohibitive, for which we employ two complementary strategies. First, we reformulate the task as a post-optimization of docking poses: an existing docking model is used to generate a coarse docking pose, which then initializes trajectory generation for refinement. Second, we employ the RL finetuning procedure in Section 3.3 to guide the generation process toward the holo state. Specifically, for a protein-ligand system with holo state $\mathbf{X}_{\text{ref}}$, the reward is defined as $r(\mathbf{X}) = -\text{rmsd}(\mathbf{X}, \mathbf{X}_{\text{ref}})$.

The training data are constructed from the PDBBind2020 database, with 23 systems reserved for testing, as detailed in Section D. With the data prepared, we perform RL finetuning using the algorithm in Algorithm 1, initializing the model with weights from the version trained on MISATO. For evaluation, we begin with the initial state constructed by docking the ligand into the apo protein

structure via AutoDock Vina (Eberhardt et al., 2021), and generate trajectories of length 100 (8 ns). The predicted holo state is identified as the conformation with the highest binding affinity, which is estimated by EPT (Jiao et al., 2024). Following Zhang et al. (2025), we report the root mean square error of the ligand (Ligand RMSD) and the pocket (Pocket RMSD) as evaluation metrics.

The evaluation results on the test set of PDBBind is shown in Table 4. Firstly, the holo state predicted by PVB without RL finetuning deviates further from the reference than the initial state, likely due to insufficient sampling from the vast conformational space within the short simulation window. Meanwhile, RL finetuning delivers substantial gains over the non-RL model across all metrics and consistently surpasses Vina-docked poses in ligand placement, suggesting that it learns to bypass local free-energy minima and progress more efficiently toward the holo state through RL.

Table 4: Results on the test set of PDBBind.

| MODELS | Ligand RMSD | | | | | Pocket RMSD | | | | | |
|---|---|---|---|---|---|---|---|---|---|---|---|
| | 25% | 50% | 75% | mean | <5Å | 25% | 50% | 75% | mean | <2Å | <4Å |
| AutoDock Vina | 4.771 | 6.268 | 8.479 | 6.368 | 26.1% | 0.707 | 1.021 | 1.380 | 1.211 | 91.3% | 95.7% |
| PVB w/o RL | 4.797 | 6.613 | 9.515 | 6.904 | 26.1% | 2.106 | 2.683 | 3.121 | 2.718 | 26.1% | 87.0% |
| PVB w/ RL | 4.178 | 5.745 | 7.500 | 5.918 | 43.5% | 1.515 | 1.962 | 2.281 | 1.965 | 56.5% | 95.7% |

In addition, the predicted and co-crystal structures of PDB 6d15 and PDB 6j0g are displayed in Figure 5, as two representative protein-ligand complex systems. Regardless of whether the initial Vina-docked pose is close to the ground truth (*e.g.*, PDB 6d15) or far from it (*e.g.*, PDB 6j0g), PVB with RL finetuning consistently evolves toward the correct binding pose within a short simulation window, demonstrating the capability to perceive global interactions rather than local ones. Therefore, we claim that PVB can serve as an efficient tool for post-optimization of docking poses.

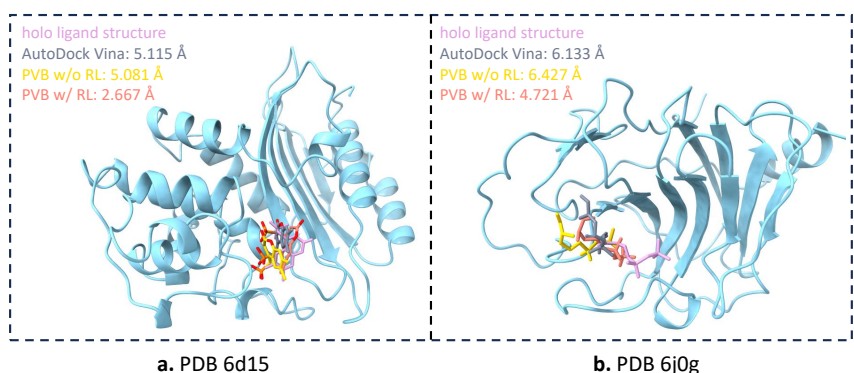

**a.** PDB 6d15        **b.** PDB 6j0g

Figure 5: Comparison of the predicted holo state with the co-crystal structure for **a.** PDB 6d15 and **b.** PDB 6j0g. The holo protein structure is shown in blue, and the displayed ligand pose is presented after aligning the predicted protein to the holo structure, with the ligand transformed accordingly.

## 5 CONCLUSION AND FUTURE WORKS

In this work, we introduce the Pretrained Variational Bridge (PVB), a novel generative model for trajectory generation for cross-domain biomolecular systems. By leveraging the encoder-decoder architecture, PVB provides a unified framework for both pretraining on single-structure data and finetuning on MD trajectories, thereby enabling seamless transfer of rich structural knowledge across stages. Extensive experiments show that, with the unified training objective, the model reproduces thermodynamic and kinetic observables consistent with classical MD and achieves substantial gains in generative fidelity over baselines. Furthermore, to facilitate efficient exploration of protein-ligand co-crystal structures, we integrate RL finetuning via adjoint matching, employing a reward function to reweigh the generative distribution in favor of the holo state. Benchmarking on PDBBind suggests PVB as an effective tool for post-optimization of docking poses.

Looking ahead, sequential trajectory generation remains a limitation despite the coarsened timestep, and developing parallelized methods that maintain temporal correlations is an important avenue for future work.

ETHICS STATEMENT

This study introduces a generative model for the simulation of cross-domain biomolecular systems. The work is primarily methodological and computational, with an emphasis on assessing the model's capacity to accurately reproduce physical observables and efficiently explore metastable states of the molecular system. All experiments were conducted *in silico* using publicly available databases or author-curated datasets. As the study does not involve human or animal subjects, clinical data, or primary biological experiments, no specific ethical approval was required.

ACKNOWLEDGMENTS

This work is jointly supported by the Fundamental and Interdisciplinary Disciplines Breakthrough Plan of the Ministry of Education of China (No. JYB2025XDXM101), the National Natural Science Foundation of China (No. 62236011, No. 62376276), and Beijing Nova Program (20230484278).

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

## APPENDIX

## A  REPRODUCIBILITY

Our code is available at `https://github.com/yaledeus/PVB`, which provides detailed instructions for training and evaluation, along with released model weights for both pretraining and downstream tasks.

## B  PROOF OF PROPOSITION

### B.1  PROPOSITION 1

Given the initial state $x_0$, the unique optimal solution to Equation (2) satisfies $p_e^*(\cdot \mid \mathbf{X}_0 = x_0) = q_e(\cdot \mid \mathbf{X}_0 = x_0)$ almost surely. Further, according to De Bortoli et al. (2023), the path measure $\mathbb{P}^*$ associated with Equation (6) preserves the coupling, *i.e.*, $\Pi_{0,1} = \mathbb{P}_{0,1} = \mathbb{P}_{0,1}^*$. Therefore, we can derive the following equalities:

$$\mathbb{P}_0^*(\mathrm{d}y_0) = \int p_e^*(\mathrm{d}y_0 \mid x_0)\, p_{X_0}(\mathrm{d}x_0) = \int q_e(\mathrm{d}y_0 \mid x_0)\, p_{X_0}(\mathrm{d}x_0) = \mathbb{P}_0(\mathrm{d}y_0), \qquad (13)$$

hence for $\mathbb{P}_0$-almost every $y_0$,

$$\mathbb{P}_{1|0}^*(\mathrm{d}y_1 \mid y_0) = \frac{\mathbb{P}_{0,1}^*(\mathrm{d}y_0, \mathrm{d}y_1)}{\mathbb{P}_0^*(\mathrm{d}y_0)} = \frac{\Pi_{0,1}(\mathrm{d}y_0, \mathrm{d}y_1)}{\mathbb{P}_0(\mathrm{d}y_0)} = \mathbb{P}_{1|0}(\mathrm{d}y_1 \mid y_0). \qquad (14)$$

By combining Equation (1), we complete the proof. $\qquad\square$

### B.2  PROPOSITION 2

We first provide the derivation of Equation (10) from Equation (9). Given the same initial state $y_0$, we consider the following two SDEs, assumed to admit unique strong solutions on $[0, 1]$:

$$\mathrm{d}\mathbf{Y}_t = \varphi_d^*(t, \mathbf{Y}_0, \mathbf{Y}_t)\, \mathrm{d}t + \sigma\, \mathrm{d}\mathbf{B}_t,\ \mathbf{Y}_0 = y_0. \qquad (15)$$

$$\mathrm{d}\mathbf{Y}_t = (\varphi_d^*(t, \mathbf{Y}_0, \mathbf{Y}_t) + \sigma u(t, \mathbf{Y}_0, \mathbf{Y}_t))\, \mathrm{d}t + \sigma\, \mathrm{d}\mathbf{B}_t,\ \mathbf{Y}_0 = y_0. \qquad (16)$$

Let $\mathbb{P}_{|0}^*$ and $\mathbb{P}_{|0}^u$ be the probability measures of the solutions to Equation (15) and Equation (16), respectively. Since the drift terms are both adapted to filtration of $\mathbf{B}_t$, we can derive the Radon-Nikodym derivative by applying the Girsanov theorem:

$$\log \frac{\mathrm{d}\mathbb{P}_{|0}^*}{\mathrm{d}\mathbb{P}_{|0}^u}(\mathbf{Y}_{0:1}) = -\int_0^1 u(t, \mathbf{Y}_0, \mathbf{Y}_t)\, \mathrm{d}\mathbf{B}_t - \frac{1}{2}\int_0^1 \|u(t, \mathbf{Y}_0, \mathbf{Y}_t)\|^2\, \mathrm{d}t,\ \mathbf{Y}_{0:1} \sim \mathbb{P}_{|0}^u. \qquad (17)$$

Accordingly, the KL divergence can be simplified by:

$$D_{\text{KL}}(\mathbb{P}^u_{|0}\|\mathbb{P}^*_{|0}) = \mathbb{E}_{\mathbf{Y}_{0:1}\sim\mathbb{P}^u}[\log\frac{\mathrm{d}\mathbb{P}^u_{|0}}{\mathrm{d}\mathbb{P}^*_{|0}}(\mathbf{Y}_{0:1}) \mid \mathbf{Y}_0 = y_0] \tag{18}$$

$$= -\mathbb{E}_{\mathbf{Y}_{0:1}\sim\mathbb{P}^u}[\log\frac{\mathrm{d}\mathbb{P}^*_{|0}}{\mathrm{d}\mathbb{P}^u_{|0}}(\mathbf{Y}_{0:1}) \mid \mathbf{Y}_0 = y_0] \tag{19}$$

$$= \mathbb{E}_{\mathbf{Y}_{0:1}\sim\mathbb{P}^u_{|0}}[\int_0^1 u(t,y_0,\mathbf{Y}_t)\,\mathrm{d}\mathbf{B}_t + \frac{1}{2}\int_0^1 \|u(t,y_0,\mathbf{Y}_t)\|^2\,\mathrm{d}t] \tag{20}$$

$$= \mathbb{E}_{\mathbf{Y}_{0:1}\sim\mathbb{P}^u_{|0}}[\frac{1}{2}\int_0^1 \|u(t,y_0,\mathbf{Y}_t)\|^2\,\mathrm{d}t]. \tag{21}$$

The last equality follows from the property that stochastic integrals are martingales. By substituting this equality into Equation (9), we obtain Equation (10).

Next, we consider the optimization problem of the inner expectation conditioned on $y_0$ at $t = 0$ in Equation (10):

$$\max_u \mathbb{E}_{\mathbf{Y}_{0:1}\sim\mathbb{P}^u}[r(\mathbf{Y}_1) - \frac{\beta}{2}\cdot\int_0^1 \|u(t,\mathbf{Y}_0,\mathbf{Y}_t)\|^2\,\mathrm{d}t \mid \mathbf{Y}_0 = y_0]. \tag{22}$$

Since $\mathbf{Y}_0 = y_0$ is fixed, the conditioning of the control vector field $u$ on $\mathbf{Y}_0$ is absorbed, yielding $u_{y_0}(t,\mathbf{Y}_t) := u(t,y_0,\mathbf{Y}_t)$. Therefore, the optimization problem of Equation (22) can be interpreted as a Stochastic Optimal Control (SOC) problem, with its standard form given by:

$$\min_u \mathbb{E}[\int_0^1 \frac{1}{2}\|u_{y_0}(t,\mathbf{Y}_t)\|^2\,\mathrm{d}t + g(\mathbf{Y}_1)], \tag{23}$$

$$\text{s.t. } \mathrm{d}\mathbf{Y}_t = (b(t,\mathbf{Y}_t) + \sigma u_{y_0}(t,\mathbf{Y}_t))\,\mathrm{d}t + \sigma\,\mathrm{d}\mathbf{B}_t, \ \mathbf{Y}_0 \sim \delta_{y_0}, \tag{24}$$

where $b(t,\mathbf{Y}_t) = \varphi^*_d(t,\mathbf{Y}_0,\mathbf{Y}_t)$, $g(\mathbf{Y}_1) = -\frac{1}{\beta}r(\mathbf{Y}_1)$ in our case, and $\delta_x$ represents the Dirac delta measure centered on $x$. Based on the theorem proposed by Domingo-Enrich et al. (2024), the optimal control $u^*_{y_0}$ of the SOC problem defined in Equation (23) is the unique critical point of the following objective:

$$\min_u \mathbb{E}_{\mathbf{Y}_{0:1}\sim\mathbb{P}^{\bar{u}}}[\frac{1}{2}\int_0^1 \|u_{y_0}(t,\mathbf{Y}_t) + \sigma\tilde{a}(t,\mathbf{Y}_{0:1})\|^2\,\mathrm{d}t \mid \mathbf{Y}_0 = y_0], \ \bar{u} = \text{sg}(u), \tag{25}$$

$$\text{s.t. } \frac{\mathrm{d}}{\mathrm{d}t}\tilde{a}(t,\mathbf{Y}_{0:1}) = -\tilde{a}(t,\mathbf{Y}_{0:1})^\top\nabla_{\mathbf{Y}_t}b(t,\mathbf{Y}_t), \ \tilde{a}(1,\mathbf{Y}_{0:1}) = \nabla_{\mathbf{Y}_1}g(\mathbf{Y}_1). \tag{26}$$

Assume that the function class for $u$ is sufficiently expressive. Then there exists a function $u^*(t,y_0,\mathbf{Y}_t)$ such that, for every $y_0 \in \text{supp}(\Pi_0)$, the map $\mathbf{Y}_t \mapsto u^*(t,y_0,\mathbf{Y}_t)$ coincides with the conditional minimizer $u^*_{y_0}(t,\mathbf{Y}_t)$. We further prove that $u^*$ is the unique minimizer of Equation (11), which can be reformulated as:

$$\mathcal{L}_{\text{adj}}(u) = \mathbb{E}_{\mathbf{Y}_0}[R_{y_0}(u_{y_0})] = \int R_{y_0}(u_{y_0})\Pi_0(\mathrm{d}y_0), \tag{27}$$

$$\text{s.t. } R_{y_0}(u) = \mathbb{E}_{t\sim\mathcal{U}(0,1),\mathbf{Y}_{0:1}\sim\mathbb{P}^{\bar{u}}}[\|u_{y_0}(t,\mathbf{Y}_t) + \sigma\tilde{a}(t,\mathbf{Y}_{0:1})\|^2 \mid \mathbf{Y}_0 = y_0], \ \bar{u} = \text{sg}(u). \tag{28}$$

By definition of $u^*_{y_0}$, for every $y_0 \in \text{supp}(\Pi_0)$ and every $u$, we have $R_{y_0}(u^*_{y_0}) \leq R_{y_0}(u)$. Integrating over $y_0$ with respect to $\Pi_0$, we have:

$$\mathcal{L}_{\text{adj}}(u^*) = \int R_{y_0}(u^*_{y_0})\Pi_0(\mathrm{d}y_0) \leq \int R_{y_0}(u)\Pi_0(\mathrm{d}y_0), \tag{29}$$

for every $u$. Thus $u^*$ is the global minimizer of $\mathcal{L}_{\text{adj}}$.

Let $u'$ for any global minimizer of $\mathcal{L}_{\text{adj}}$. Consider the set:

$$A := \{y_0 : R_{y_0}(u') > R_{y_0}(u^*_{y_0})\}. \tag{30}$$

If $\Pi_0(A) > 0$ then:

$$\mathcal{L}_{\text{adj}}(u') - \mathcal{L}_{\text{adj}}(u^*) = \int (R_{y_0}(u') - R_{y_0}(u^*_{y_0}))\Pi_0(\mathrm{d}y_0) \tag{31}$$

$$\geq \int_A (R_{y_0}(u') - R_{y_0}(u^*_{y_0}))\Pi_0(\mathrm{d}y_0) > 0. \tag{32}$$

This contradicts minimality of $u'$. Hence $\Pi_0(A) = 0$. Therefore for $\Pi_0$-almost every $y_0$ we have $R_{y_0}(u') = R_{y_0}(u^*_{y_0})$. Recall that $u^*_{y_0}$ is the unique minimizer of Equation (25), thus $u' = u^*$ almost surely w.r.t. $\Pi_0$. This completes the proof. $\qquad\square$

## C  MODEL ARCHITECTURE

### C.1  BLOCK-LEVEL REPRESENTATION

While atomic numbers offer a basic representation of molecular systems, different classes of molecules are often characterized by domain-specific building blocks (*e.g.*, amino acids in proteins). To enhance the model's ability to generalize across molecular domains, we aim to represent molecular systems in a hierarchical manner based on these basic units. Inspired by Kong et al. (2024), We constructed a vocabulary of building blocks that includes all atomic types with atomic numbers from 1 to 118, as well as the 20 natural amino acids. Based on this vocabulary, we provide the block-level representation $\boldsymbol{b} \in \mathbb{N}^N$ for the molecular systems, where $\boldsymbol{b}[i]$ ($0 \leq i < N$) is defined as:

- the atom type of the $i$-th atom, if it is part of a small molecule;
- the amino acid type containing the $i$-th atom, if it is part of a protein.

The atomic type $\boldsymbol{z}$ and the block representation $\boldsymbol{b}$ will be projected into the same latent space dimension $H$ through two separate embedding layers. The two embeddings are summed to form the initial atomic feature representation $\boldsymbol{H} \in \mathbb{R}^{N \times H}$, which is then passed to the subsequent network.

### C.2  GRAPH-LEVEL REPRESENTATION

The molecular system is further represented as a graph $\mathcal{G} = (\mathcal{V}, \mathcal{E})$, where $\mathcal{V}$ includes the invariant features $\boldsymbol{H}$ and SO(3)-equivariant Euclidean coordinates of $N$ nodes (*i.e.*, heavy atoms), and $\mathcal{E}$ comprises the edge index and associated edge features.

Specifically, for each molecule in the system, we construct intra-molecular edges using the kNN algorithm (Cover & Hart, 1967), linking each node to its $k$ nearest atoms in the molecule. In multi-molecule systems such as protein–ligand complexes, we further build inter-molecular edges by connecting each ligand atom to its $k$ nearest protein atoms. The constructed edge index is denoted by $\boldsymbol{E} \in \mathbb{N}^{2 \times E}$, where $E$ is the number of edges. Moreover, from the covalent bond index $\boldsymbol{C}$, we derive the edge feature $\boldsymbol{c} \in \{0, 1\}^E$, which explicitly encodes the molecular topological structure:

$$\boldsymbol{c}[i] = \begin{cases} 1, & \text{if the } i\text{-th edge belongs to } \boldsymbol{C}, \\ 0, & \text{otherwise.} \end{cases}, \ 0 \leq i < E. \tag{33}$$

### C.3  NEURAL NETWORK BACKBONE

With the molecular graph prepared, the well-known `TorchMD-NET` (Pelaez et al., 2024) is leveraged as the backbone model to preserve SO(3)-equivariance. We propose a simple modification in which the topology edge feature $\boldsymbol{c}$ is concatenated to the default edge attributes in the network after embedding.

In particular, to allow the decoder $\varphi_d$ to condition on both $\mathbf{Y}_0$ and $\mathbf{Y}_t$, we first construct the graph $\mathcal{G}$ from $\mathbf{Y}_t$. Next, the equivariant coordinates of $\mathbf{Y}_0, \mathbf{Y}_t$ are converted into invariant interatomic distances as edge features, and processed by a lightweight equivariant graph neural network to obtain invariant node features of dimension $H$. These features are concatenated, fused through an MLP with a single hidden layer, and then fed into the subsequent attention layers. Finally, the Geometric Vector Perceptron (GVP) used in (Yu et al., 2025) is applied as the output layer to yield invariant node features and SO(3)-equivariant vectors.

# D    DATASET DETAILS

Firstly, the following datasets are used for pretraining:

- **PCQM4Mv2** (Hu et al., 2021). A quantum chemistry dataset comprising approximately 3M small molecules, with DFT-calculated 3D structures.

- **ANI-1x** (Smith et al., 2020). A dataset with 5M small organic molecules, consisting of multiple QM properties from density functional theory calculations.

- **PDB** (Berman et al., 2000). A large collection of experimentally determined 3D structures of biological macromolecules, where we use a subset processed by Jiao et al. (2024), comprising 58,576 protein monomers for training and 1,477 for validation.

- **PDBBind2020** (Wang et al., 2004; 2005; Liu et al., 2015). A curated collection of 19,443 crystal protein-ligand complex structures with experimentally measured binding affinities. In this work, we adopt the data split proposed by (Lu et al., 2024), which consists of 12,826 complexes for training, 734 for validation, and 334 for testing.

For datasets without a pre-defined split, we randomly partition them into training and validation sets with a 4:1 ratio.

Next, we benchmark the trajectory generation task on proteins and protein-ligand complexes. We use the following three datasets for the trajectory generation task:

- **ATLAS** (Vander Meersche et al., 2024). The ATLAS dataset provides long-timescale trajectory data for thousands of proteins generated from 100 ns all-atom MD simulations performed with GROMACS (Abraham et al., 2015) in explicit TIP3P water (Jorgensen et al., 1983). Following Yu et al. (2025), we use the curated subset comprising 790 proteins for training and 14 for evaluation, where the split is constructed by clustering sequences with MMseqs2 (Steinegger & Söding, 2017) under a 30% sequence-identity threshold to ensure non-redundancy between the training and test sets.

- **mdCATH** (Mirarchi et al., 2024). The dataset contains all-atom molecular systems for 5,398 CATH domains, each simulated using a modern classical force field. For every domain, five independent MD trajectories are generated at each of five temperatures ranging from 320 K to 450 K, with the frame spacing of 1 ns. We follow the data split proposed by Janson et al. (2025), comprising 5,049/40/90 protein domains for the training, validation, and test sets, respectively. To reduce the computational cost, we use only the trajectories simulated at 320 K for training and evaluation. Training data pairs $(x_t, x_{t+\tau})$ are sampled every 5 ns from MD trajectories, with the coarsened time step $\tau = 1$ ns, yielding approximately 1.5M training pairs and 18K validation pairs. The first 20 domains of the original test set are used as our evaluation subset.

- **MISATO** (Siebenmorgen et al., 2024). The MISATO dataset consists of over 10,000 protein–ligand complexes, each accompanied by a 10 ns explicit-solvent molecular dynamics trajectory. Following the original protocol, the first 2 ns of each trajectory are discarded, and the remaining 8 ns are provided as equilibrated conformational ensembles. In our experiments, we adopt the original dataset split of 13,765/1,595/1,612 for training/validation/test, respectively, while restricting the evaluation to the first 20 test complexes due to the substantial computational cost of trajectory generation on large systems.

To explore the protein-ligand holo state, we benchmark on the PDBBind2020 database (Wang et al., 2004; 2005; Liu et al., 2015). To prepare data for RL finetuning and ensure the availability of apo protein and unbound ligand structures for evaluation, we apply the following pre-processing pipeline to the training and test sets, respectively.

- **PDBBind-train**. Given the sensitivity of protein-ligand interactions to structural accuracy, we use only the refined set of PDBBind2020, which contains 5,316 diverse protein-ligand complexes with high-resolution crystal structures. To prevent sequence-level overlap, we first cluster complexes using MMSeqs2 (Steinegger & Söding, 2017) based on their binding pocket residue sequences with a 60% sequence identity threshold, retaining a single representative from each cluster. Next, we adopt the time-based split used in previous

studies (Corso et al., 2022; Stärk et al., 2022; Lu et al., 2024), where structures deposited before 2019 are randomly partitioned into training and validation sets in a 4:1 ratio, and those deposited in 2019 are held out for testing. Furthermore, since RL finetuning requires not only crystal structures but also binding intermediates of protein-ligand complexes, we select a small subset from the training and validation sets for additional MD simulations, yielding a 265/70 split. For each selected system, a short 20 ps MD simulation is performed using `OpenMM` (Eastman et al., 2023) in the NVT ensemble at 300 K. The AMBER14 force field (Maier et al., 2015) is applied, with explicit TIP3P water (Jorgensen et al., 1983) and 1.0 nm solvent padding. Long-range electrostatics are treated with PME, and all hydrogen bond lengths are constrained. The system is integrated using `LangevinMiddleIntegrator` with a friction coefficient of $0.5 \, \text{ps}^{-1}$ and a 2 fs timestep. Finally, 10 snapshots are extracted from each trajectory at 2 ps spacing for RL finetuning.

- **PDBBind-test**. For the test set, we first remove complexes that contain non-canonical amino acids, and then exclude those overlapping with the training and validation sets of MISATO, leaving a total of 33 test cases. For each test case, the unbound ligand and apo protein structures are generated using MMFF94 (Halgren, 1996) and Boltz-1 (Wohlwend et al., 2025), respectively. These structures are rigidly aligned to their corresponding parts of the crystal structure via the Kabsch algorithm (Kabsch, 1976). The aligned structures then serve as input for AutoDock Vina (Eberhardt et al., 2021), which predicts a docking pose of the ligand while keeping the apo protein structure fixed, using an exhaustiveness parameter of 32. After removing the cases where Vina-docked ligands fail `rdkit` (rdk) sanitization, 23 test systems are retained, where the Vina-docked complex structures will be used as the initial states for trajectory generation.

# E    TRAINING DETAILS

## E.1    POSITION PERTURBATION

Recall that our objective is to generate trajectories from the initial state of a molecular system, a process prone to error accumulation due to the inherent instability of generative models. Consequently, if the model is trained solely on reasonable conformations such as crystal structures or snapshots from MD trajectories, even minor deviations during generation may cause the model to fall out of distribution. To enhance the robustness of trajectory generation, we introduce a small perturbation to the initial state $\mathbf{X}_0$ before feeding it into the encoder during both pretraining and finetuning. Formally,

$$\mathbf{X}_0 \leftarrow \mathbf{X}_0 + \sigma_p \boldsymbol{\epsilon}, \ \boldsymbol{\epsilon} \sim \mathcal{N}(\mathbf{0}, \boldsymbol{I}), \tag{34}$$

where $\sigma_p$ is the hyperparameter of perturbation strength.

## E.2    DATA AUGMENTATION FOR RL

Since the training data for reinforcement learning are limited and sampled only from short MD trajectories, we apply data augmentation during RL finetuning. Specifically, for each training iteration, we randomly sample a rotation angle $\theta \sim \mathcal{U}(-\frac{\pi}{3}, \frac{\pi}{3})$ and a unit rotation axis, which can be transformed to a random rotation matrix $\boldsymbol{Q} \in \text{SO}(3)$ using Rodrigues' rotation formula. Next, we sample a translation vector $\boldsymbol{t}$ from the standard Gaussian distribution. The $\text{SE}(3)$-transformation $(\boldsymbol{Q}, \boldsymbol{t})$ is then applied to the ligand pose of the initial state $\mathbf{X}_0$, which served as the new input.

## E.3    ALGORITHM FOR RL FINETUNING

We present the RL finetuning algorithm for protein-ligand complexes in Algorithm 1, where the operator data_augmentation is discussed in the previous section.

---

**Algorithm 1** RL finetuning for protein-ligand complexes with PVB

---

1: **Input:** a protein-ligand complex system with position $x_0$, the corresponding holo state $x_{\text{ref}}$, the reference models $\varphi_e, \varphi_d$, the regularization strength $\beta$, discretized SDE steps $T$.
2: freeze parameters of $\varphi_e$ and $\varphi_d$
3: $\varphi_d^u \leftarrow \varphi_d$
4: $\Delta \leftarrow \frac{1}{T}$
5: **for** training iterations **do**
6:     $x_0 \leftarrow \text{data\_augmentation}(x_0)$
7:     $t \sim \mathcal{U}(\{0, \Delta, 2\Delta, \cdots, 1\})$
8:     $y_0 \sim p_e(\cdot \mid \mathbf{X}_0 = x_0)$ {reparameterization}
9:     $\mathbf{Y}_{0:1} \coloneqq [\mathbf{Y}_0, \mathbf{Y}_\Delta, \mathbf{Y}_{2\Delta}, \cdots, \mathbf{Y}_1] \sim p_d^u(\cdot \mid \mathbf{Y}_0 = y_0)$ {Equation (8)}
10:     $\tilde{a}_1 \leftarrow -\frac{1}{\beta} \nabla_{\mathbf{Y}_1} \text{rmsd}(\mathbf{Y}_1, x_{\text{ref}})$
11:     $s \leftarrow 1$
12:     **while** $s > t$ **do**
13:         $\tilde{a}_{s-\Delta} = \tilde{a}_s + \tilde{a}_s^\top \nabla_{\mathbf{Y}_s} \varphi_d(s, \mathbf{Y}_0, \mathbf{Y}_s) \cdot \Delta$
14:         $s \leftarrow s - \Delta$
15:     **end while**
16:     $\mathcal{L}_{\text{adj}} \leftarrow \|\varphi_d^u(t, \mathbf{Y}_0, \mathbf{Y}_t) - \varphi_d(t, \mathbf{Y}_0, \mathbf{Y}_t) + \sigma^2 \tilde{a}_t\|^2$
17:     $\min \mathcal{L}_{\text{adj}}$
18: **end for**

---

### E.4 HYPERPARAMETERS

The hyperparameters used for training and inference with PVB are summarized in Table 5.

Table 5: Hyperparameters of PVB.

| Hyperparameters | Values |
|---|---|
| Hidden dimension $H$ | 256 |
| FFN dimension | 512 |
| RBF dimension | 32 |
| RBF cutoff | 10 Å |
| # Attention heads | 8 |
| # Encoder layers | 2 |
| # Decoder layers | 8 |
| # Neighbors $k$ | 32 |
| Variational noise scale $\sigma_e$ | $\sqrt{0.5}$ Å |
| SDE noise scale $\sigma$ | 0.2 Å |
| Perturbation Strength $\sigma_p$ | 0.2 Å |
| KL loss weight $w_{\text{KL}}$ | 0.8 |
| ABM loss weight $w_{\text{ABM}}$ | 1.0 |
| Regularization strength $\beta$ | 4e-4 |
| Learning rate (pretrain) | 1e-4 |
| Learning rate (finetune) | 5e-5 |
| Optimizer | Adam |
| Warm up steps | 1,000 |
| Warm up scheduler | LambdaLR |
| Training scheduler | ReduceLRonPlateau(factor=0.8, patience=5, min\_lr=1e-7) |
| Discretized SDE steps $T$ | [8,10] |

# F    EVALUATION DETAILS

## F.1    FEATURIZATION OF PROTEINS

To featurize trajectories of proteins in ATLAS, we select as collective variables the sine and cosine of backbone $\phi$ and $\psi$ torsions and all side-chain torsions, and the Euclidean distances of contacting heavy-atom pairs, which are defined as pairs of C, N, or S atoms separated by more than two bonds in the molecular topology graph and lying within 4.5 Å in the initial conformation of the MD reference trajectory (Yuan et al., 2012). In particular, since each trajectory in mdCATH contains relatively few frames, we instead use only the backbone $\phi$ and $\psi$ torsions as TICA features for this dataset to avoid ill-conditioning. The extracted features are then projected onto the slowest two time-lagged independent components (TIC0 and TIC1) (Pérez-Hernández et al., 2013) using `deeptime` (Hoffmann et al., 2021), with a lag time of 100 ps for ATLAS and 1 ns for mdCATH.

## F.2    JENSEN-SHANNON DIVERGENCE

We compute the Jensen-Shannon Divergence (JSD) on various feature spaces using `scipy` (Jones et al., 2001). For 1-dimensional JSD, we discretize the range spanning the minimum and maximum values from MD trajectories into 50 bins. For multi-dimensional features, we first compute the JSD separately along each dimension and then report the mean value across all dimensions.

## F.3    DECORRELATION

Following (Jing et al., 2024b), we report the decorrelation percentage for TIC0 to characterize the kinetic properties of the generated trajectories. Firstly, we define the autocorrelation as follows:

$$\mathbb{E}[(y_t - \mu)(y_{t+\Delta t} - \mu)]/\sigma^2, \tag{35}$$

where $\mu$ and $\sigma$ are the mean and standard values of TIC0 from the MD reference trajectory, respectively. By computing the autocorrelation using `statsmodels` (Seabold & Perktold, 2010), the generated trajectory is defined as *decorrelated* if its autocorrelation of TIC0 starts above and falls below 0.5 within 1,000 frames (100 ns). Since all MD reference trajectories have been verified to decorrelate within 100 ns, the decorrelation fraction observed in the generated trajectories (Decorr-TIC0) serves as a quantitative indicator of the ability to capture dominant slow modes during simulations.

## F.4    MARKOV STATE MODEL

We employ Markov state models to assess the fidelity of metastable state reproduction. We first cluster the two-dimensional features (TIC0 and TIC1) from the MD reference trajectory into 100 microstates using the KMeans++ algorithm (Arthur & Vassilvitskii, 2006). Next, we construct the Markov state model based on the microstates with a lag time of 100 ps, and apply the PCCA+ algorithm (Röblitz & Weber, 2013) to coarse-grain the dynamics into 10 metastable states. Specifically, after the PCCA+ analysis, we reassign clusters corresponding to inactive microstates to the nearest active cluster, thereby ensuring connectivity among all microstates. Finally, the clustering and PCCA models are employed to assign metastable states for each frame of both the MD reference trajectory and the generated trajectory, and JSD is computed between their probability distributions over the 10 metastable states.

## F.5    ALIGNMENT OF PROTEIN-LIGAND COMPLEXES

To assess the protein-ligand complexes, the frames should be superimposed onto a reference conformation at first. Specifically, for the PDBBind dataset, the holo state is used as the reference conformation, whereas for the MISATO dataset, the initial state of the MD reference trajectory is selected. Given the reference conformation, the pocket of the complex is defined as the set of residues containing at least one heavy atom within 20 Å of the ligand center. Each frame in the trajectory is then aligned to the reference conformation by applying an SE(3)-transformation based on the backbone C$\alpha$, C, and N backbone atoms of residues in the pocket.

### F.6 CONTACT OCCUPANCY

Following Janson et al. (2023), for a protein with $R$ residues, we define the contact occupancy $r(i, j)$ of the residue pair $i, j$ as:

$$r(i, j) = \frac{1}{T} \sum_{t=1}^{T} \mathbf{1}_{d_t(i,j)<10\text{Å}}, \ 1 \leq i < j \leq R, \tag{36}$$

where $T$ denotes the number of frames in the trajectory, and $d_t(i, j)$ denotes the Euclidean distance between C$\alpha$ atoms of residues $i$ and $j$ in the $t$-th frame. Then the root mean square error of the contact occupancy is computed by:

$$\text{RMSE-CONTACT} = \sqrt{\frac{2}{R(R-1)} \sum_{1 \leq i < j \leq R} (r(i, j) - r_{\text{ref}}(i, j))^2}, \tag{37}$$

where $r_{\text{ref}}(i, j)$ is calculated from the MD reference trajectory.

Similarly, for protein-ligand complexes, we select the heavy-atom pairs between the ligand and the pocket to calculate contact occupancy, and correspondingly compute the RMSE relative to the MD reference trajectory.

### F.7 VALIDITY

For the molecular system of proteins, we calculate the validity of generated conformations to assess the stability in long timescale simulations. Following Wang et al. (2024a), a conformation is defined as *valid* if: (I) the Euclidean distance between C$\alpha$ atoms of all residue pairs is no less than 3.0 Å and (II) the Euclidean distance between C$\alpha$ atoms of adjacent residue pairs is no more than 4.19 Å. The proportion of valid conformations in the trajectory is reported, denoted as VAL-CA.

## G ADDITIONAL EXPERIMENTAL RESULTS

### G.1 FAST-FOLDING PROTEINS

In this section, we assess whether PVB can explore essential biological processes in long-timescale MD simulations. As a case study, we focus on the well-established fast-folding proteins, which capture protein folding and unfolding dynamics on the microsecond timescale (Lindorff-Larsen et al., 2011). Among twelve fast-folders, we select NTL9 (an $\alpha/\beta$ protein), Protein B (a three-helix bundle protein), Villin (an actin-associated mini-protein), the WW domain (a $\beta$-sheet protein), and $\lambda$-repressor (a helix-turn-helix DNA-binding protein) as five representative test systems, with reference trajectories obtained from MD simulations ranging from 100 to 3000 $\mu$s, saved at a frame spacing of 200 ps.

To better characterize folding dynamics, we employ global structural descriptors as TICA collective variables: the sine and cosine of backbone ($\phi$, $\psi$) and side-chain torsions, along with C$\alpha$-C$\alpha$ distances for residue pairs separated by at least two residues. The lag time for TICA is taken as 20 ns.

Using the PVB model trained on ATLAS, we generate trajectories of length 100,000 for the five test systems in a zero-shot manner, each corresponding to 10 $\mu$s in wall-clock time. The results are visualized in Figure 6. Here, contact occupancy is defined as the fraction of residue pairs separated by at least two residues, whose C$\alpha$ atoms are within 12 Å (Kukic et al., 2014). We observe that for NTL9, PVB fluctuates mainly around the initial conformation during the 10 $\mu$s simulations without reaching the folded state, which is likely due to the limited simulation timescale and the system being partly out of distribution for the model. For Protein B, although the limited simulation time precludes a full recreation of MD findings, PVB nonetheless provides an initial characterization of metastable states consistent with MD. Moreover, for Villin, the WW domain, and $\lambda$-repressor, despite minor artifacts in phase space regions undersampled by MD (*e.g.*, for $\lambda$-repressor), the model nevertheless recovers the essential free energy landscape and captures the rapid transition between metastable states, highlighting its capacity to generalize to reliable long-timescale molecular dynamics of fast-folding proteins spanning diverse secondary structures.

Regarding contact occupancy, PVB demonstrates robust stability for NTL9, Protein B, Villin, and the WW domain throughout the 10 $\mu$s simulations, with trajectories displaying average values and fluctuations broadly consistent with the MD benchmarks. In contrast, the $\lambda$-repressor case reveals a declining trend, signaling simulation failure and subsequent structural collapse. These findings underscore PVB's capacity to maintain long-timescale fidelity for small- to medium-sized proteins, with potential challenges in scaling to larger systems.

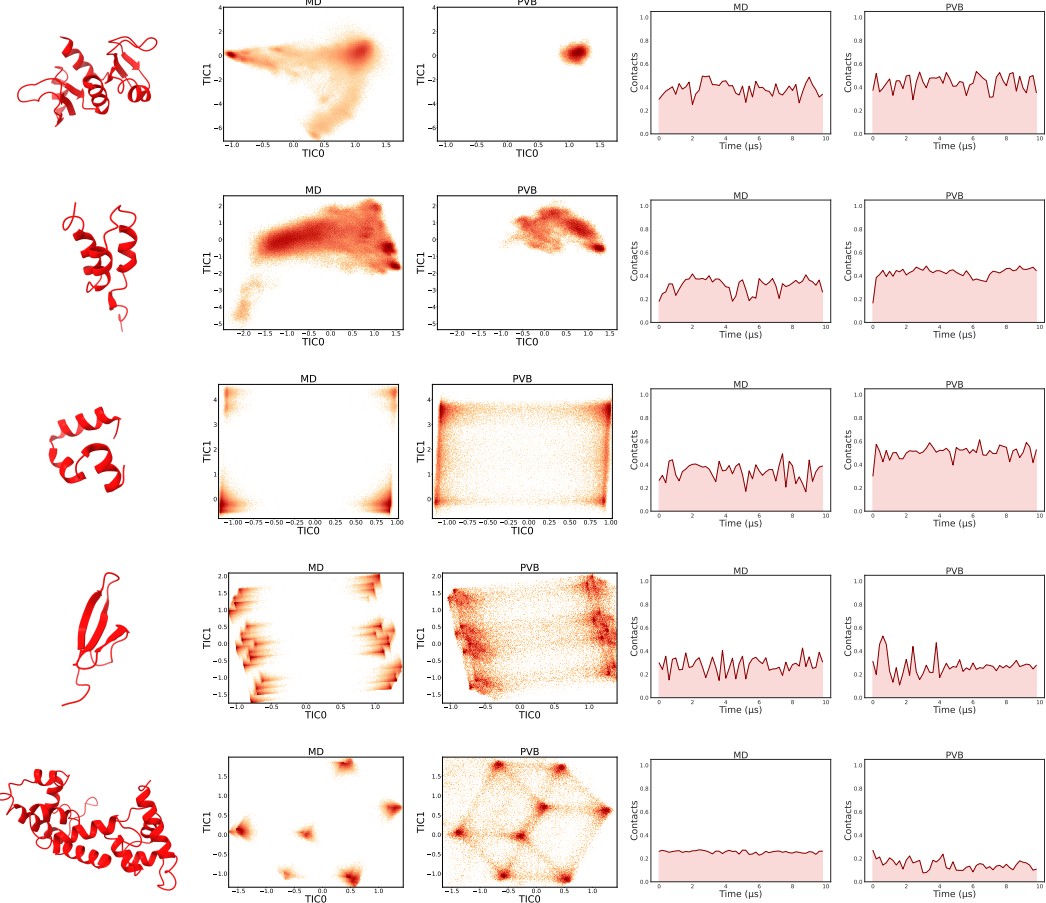

Figure 6: Visualization of the generated trajectories of NTL9 (**Row 1**), Protein B (**Row 2**), Villin (**Row 3**), the WW domain (**Row 4**), and $\lambda$-repressor (**Row 5**). **Column 1**: Experimental determined structures from Protein Data Bank (PDB). **Column 2-3**: Comparison of free energy surfaces projected onto the slowest two time-lagged independent components (*i.e.*, TIC0 and TIC1) for the entire MD reference and PVB-generated trajectories. **Column 4-5**: Comparison of the contact occupancy during the first 10 $\mu$s of simulation for the MD reference and PVB-generated trajectories.

## G.2    ABLATION STUDY

To validate the effectiveness of the proposed pretrained variational bridge in leveraging structural prior knowledge, we perform the following ablation experiments on the ATLAS dataset under three different configurations:

- **PVB w/o finetuning**. This setting corresponds to directly using the pretrained model to generate trajectories on the test systems, without any finetuning on the training data.

- **PVB w/o pretraining**. This setting corresponds to training the model from scratch on the ATLAS training set, without using any pretrained weights.

- **PVB-ITO**. This setting replaces our bridge-matching-based decoder with the conditional diffusion model used in ITO (Schreiner et al., 2023), while keeping the model backbone unchanged.

The evaluation results are presented in Table 6. First, they show that PVB with pretraining achieves substantial improvements across all metrics, with particularly pronounced gains in validity. This underscores the critical role of the rich, cross-domain structural knowledge acquired during pretraining.

Second, directly evaluating the pretrained PVB model on ATLAS without finetuning reveals that the fidelity of generated conformations largely stem from pretraining. While finetuning slightly reduces residue contacts, it consistently improves distributional similarity, as reflected by lower JSD values. Together, these findings provide strong evidence that PVB functions effectively as a unified framework, benefiting from both pretraining and finetuning.

Finally, we also tested the scenario where the PVB decoder is entirely replaced with the conditional diffusion model used in ITO, while fully replicating the PVB pretraining and finetuning scheme. The experimental results show that the modified model performs poorly overall and fails to generate physically plausible protein conformations. This suggests, to some extent, the limitations and incompatibility of ITO's generative framework in this setting. We also emphasize that, although bridge matching is not necessarily irreplaceable, it does perform remarkably well under the current experimental setup.

Table 6: Ablation results on the ATLAS test set, with each metric reported as mean/std across 14 test protein monomers. The best result for each metric is shown in **bold**.

| MODELS | JSD (↓) | | | | VAL-CA (↑) | RMSE-CONTACT (↓) |
|---|---|---|---|---|---|---|
| | Rg | Torus | TIC | MSM | | |
| PVB | **0.457**/0.087 | 0.336/0.030 | **0.371**/0.087 | **0.333**/0.105 | **0.975**/0.023 | 0.143/0.031 |
| PVB w/o finetuning | 0.561/0.123 | **0.310**/0.018 | 0.463/0.094 | 0.407/0.155 | 0.968/0.039 | **0.097**/0.014 |
| PVB w/o pretraining | 0.710/0.063 | 0.481/0.011 | 0.408/0.072 | 0.581/0.127 | 0.022/0.007 | 0.223/0.033 |
| PVB-ITO | 0.676/0.082 | 0.577/0.014 | 0.487/0.072 | 0.460/0.158 | 0.000/0.000 | 0.269/0.042 |

## G.3 PHYSICALITY OF GENERATED PROTEIN-LIGAND COMPLEXES

In the docking pose post-optimization task for protein-ligand complexes (Section 4.2), we relied solely on RMSD as the evaluation metric, which may obscure the extent to which the generated conformations are physically plausible. In this section, we introduce more fine-grained evaluation criteria to analyze the physicality of both protein and ligand conformations produced along the generated trajectories, and further compare how it changes before and after reinforcement learning.

Specifically, to evaluate the physicality of the generated complexes, we introduce the following metrics:

- **VAL-BL**. A ligand conformation is *bond-length valid* if every covalent bond length deviates by no more than $20\%$ from the Distance Geometry (DG) reference bounds defined in `PoseBusters` (Buttenschoen et al., 2024).

- **VAL-BA**. A ligand conformation is *bond-angle valid* if every bond angle stays within $20\%$ of the DG reference angular bounds defined in `PoseBusters`.

- **SCF**. A ligand conformation is *steric clash-free* if all non-covalent inter-atomic distances are at least $80\%$ of their DG lower bounds defined in `PoseBusters`.

- **SI**. A ligand structure satisfies *stereochemical integrity* only if (i) all topologically stereogenic atoms possess well-defined CIP configurations, and (ii) all stereogenic double bonds exhibit valid E/Z annotations without ambiguity. The analyses are carried out using `rdkit` (rdk).

- **VAL-CA**. To assess the validity of protein conformations, we adopt the same metric defined in Section F, considering a protein structure *valid* only if there are no broken bonds between adjacent residues and no clashes between any residue pairs based on C$\alpha$ distances.

Note that for all the above metrics, we first compute each metric over the 100-frame generated trajectories for each test system and take the average. We then report the mean and standard deviation of these averages across all 23 test systems, which are shown in Table 7. Based on the comparison of the experimental results, we draw the following observations.

First, the model trained on MISATO (*i.e.*, PVB w/o RL) preserves a high degree of physical plausibility in the generated all-atom conformations of both ligands and proteins. This further demonstrates that large-scale pretraining on diverse single structures, followed by light MD finetuning, effectively imparts domain knowledge applicable across molecular families.

Second, after reinforcement learning, the model exhibits slight improvements on ligand-related metrics and a notable enhancement in the validity of generated protein conformations. This indicates that reinforcement learning preserves, and may even enhance, the model's ability to generate physically plausible trajectories supported by additional data.

Finally, we find that the two models perform identically on the stereochemical integrity metric. Upon inspection, we determined that in 3 of the 23 test systems, the initial ligand conformations from the original database already failed this test. Since the input specifies the molecular topology, the generated trajectories do not alter stereochemistry of the ligand, leading all ligand conformations in the trajectories to fail the check as well. This accounts for the unusual consistency observed for this metric.

Table 7: Physicality of the generated protein-ligand complex trajectories on the test set of PDBBind. The metrics are reported in mean/std over 23 test systems.

| MODELS | VAL-BL ($\uparrow$) | VAL-BA ($\uparrow$) | VAL-CA ($\uparrow$) | SCF ($\uparrow$) | SI ($\uparrow$) |
|---|---|---|---|---|---|
| PVB w/o RL | 0.874/0.200 | 0.827/0.212 | 0.615/0.217 | 0.962/0.051 | 0.870/0.344 |
| PVB w/ RL | 0.895/0.204 | 0.852/0.220 | 0.813/0.224 | 0.963/0.070 | 0.870/0.344 |

### G.4 INFERENCE EFFICIENCY

We assess the inference efficiency for different models by computing the inference time per step on the same GPU device. Results are shown in Figure 7, where PVB delivers approximately 5-10$\times$ faster inference than the second-best model (MDGEN), while also exhibiting the lowest variance across all baselines.

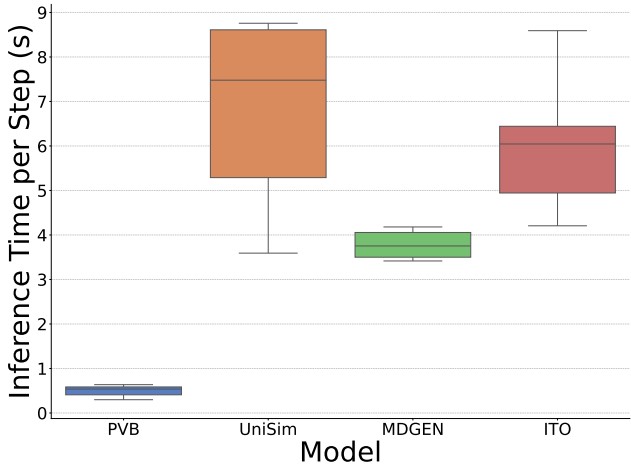

Figure 7: Comparison of per-step inference time on the ATLAS test set for deep learning models, evaluated on a single identical GPU.

## H  COMPUTING INFRASTRUCTURE

All experiments were performed on NVIDIA GeForce RTX 3090 GPUs. During the pretraining stage, we used five GPUs for approximately five days. To fully utilize GPU memory across molecular systems of varying sizes, the mini-batch size was dynamically adjusted, maintaining an average of about 2,500 atoms per batch. In addition, to balance the contribution of datasets with different sizes, each training epoch randomly samples up to 60,000 mini-batches from each dataset. For smaller datasets, all available samples are traversed within the epoch.

Following pretraining, finetuning was conducted on up to five GPUs, with training durations ranging from one day to one week depending on the specific task and dataset size.

Finally, all model evaluations, including both baseline methods and our proposed approach, were performed on a single GPU to ensure a fair and consistent comparison.

## I  DISCLOSURE OF LARGE LANGUAGE MODEL USAGE

Large language models (LLMs) were used solely as general-purpose writing assistants in the preparation of this manuscript. Specifically, LLMs were employed to polish wording, improve clarity, and correct grammar. They were not used for research ideation, methodological design, experimental planning, data analysis, or any scientific decision-making. All technical content, results, and conclusions were developed and verified by the authors, who take full responsibility for the manuscript.

