# OpenReview forum: "Unified Biomolecular Trajectory Generation via Pretrained Variational Bridge"
_ICLR.cc/2026/Conference — ICLR 2026 Poster_

### Official Review · Reviewer_BUEn · 2025-10-21

**Soundness:** 3
**Presentation:** 3
**Contribution:** 3
**Rating:** 8
**Confidence:** 4

**Summary:**

This paper introduces the Pretrained Variational Bridge (PVB), a unified generative framework for biomolecular trajectory generation that leverages pretraining on static 3D structures and finetuning on coarse-grained MD trajectory data.
The central idea is to bridge the gap between single-structure pretraining and trajectory-conditioned finetuning with a unified objective:  $\mu(X_1 \mid X_0)$.

During pretraining, the model learns from diverse molecular structures to capture cross-domain structural knowledge.
During finetuning, the model is finetuned on paired transition data $(X_t, X_{t + \Delta t})$ from dynamic datasets (e.g., ATLAS and MISATO).

An additional innovation is the RL-based adjoint finetuning using stochastic optimal control, enabling direct optimization for holo state generation in protein–ligand systems.

Empirical results show that PVB reproduces thermodynamic and kinetic observables (Rg, torsion, TIC projections, MSM occupancy) with stability comparable to MD and substantial improvement over baselines (ITO, MDGen, UniSim) in validity (VAL-CA = 0.97) and decorrelation metrics.
In protein–ligand docking, PVB with RL finetuning outperforms AutoDock Vina and non-RL variants.

**Strengths:**

* PVB elegantly integrates structural pretraining and trajectory learning through a shared encoder–decoder bridge, aligning objectives across domains. Pretraining on heterogeneous biomolecular structures allows transfer to proteins, small molecules, and protein–ligand complexes without retraining.

* The adjoint-based stochastic control formulation enables memory-efficient fine-tuning toward functional states (e.g., holo forms) without additional networks.

* PVB consistently achieves better or comparable results to classical MD and generative baselines in reproducing both kinetic and thermodynamic observables.

* The RL variant shows meaningful progress toward real drug-design applications, improving ligand placement beyond traditional docking and static generative methods, which can serve as an alternative method for docking.

**Weaknesses:**

* While conceptually elegant, the experimental scope is somewhat narrow and the demonstrated benefits are modest under realistic scales. The datasets (ATLAS, MISATO) are relatively small, and the observed improvements, though consistent, are incremental, especially given that baseline MDGen and UniSim already yield physically valid trajectories. Including results on the recently released MDCATH dataset will be make the manuscript stronger.

* As mentioned by the authors, the generation remains sequential, limiting scalability to long timescales or high-throughput ensemble generation. The paper does not analyze the runtime of the PVB for trajectory generation, which is also a concerning fact. While coarse timesteps improve efficiency conceptually, inference speed, wall-clock cost, and scaling to larger systems (e.g., >10⁴ atoms) remain unreported.

* The claimed cross-domain transferability is supported only by protein and protein–ligand tasks; other molecular domains (RNA, materials, polymers) are underexplored.

**Questions:**

1. I am particularly interested in the experimental results on the large-scale MDCath dataset, as well as the runtime analysis of the proposed method. Could the authors provide more details or quantitative comparisons to illustrate the computational efficiency and scalability of PVB?

2. Is it possible to extend the proposed framework for parallel trajectory generation, rather than sequential sampling? This could further improve scalability, especially for long biomolecular simulations.

3. Lastly, could the authors elaborate on the role of the latent variable X0? Specifically, in the statement “The latent variable X0 is introduced to avoid the collapse of the conditional probability µ(X1|X0) from degenerating into a Dirac delta measure," it would be helpful to clarify why this latent variable is necessary and what would happen if one directly generated X1 conditioned on X0 without it.

---

> ### Author Response · Authors · 2025-11-21
>
> We thank the reviewer for the careful reading and valuable suggestions. We are pleased that the reviewer found our approach and results meaningful, and we have carefully considered each of the points raised. The following responses aim to clarify misunderstandings, provide additional evidence, and highlight the contributions of our work more clearly.
>
> - **Q1**: While conceptually elegant, the experimental scope is somewhat narrow and the demonstrated benefits are modest under realistic scales. The datasets (ATLAS, MISATO) are relatively small, and the observed improvements, though consistent, are incremental, especially given that baseline MDGen and UniSim already yield physically valid trajectories.
>
> **A1**: Thank you for the thoughtful comment. We would like to highlight that the physical validity of the generated trajectories is the central challenge in this setting. As shown in **Table 1**, only about 10% of the trajectories produced by MDGen and UniSim satisfy basic physicality checks, whereas our PVB model achieves 97.3%. This makes PVB the only approach that attains the same order of physical plausibility as the MD oracle. We therefore believe that, even within the current experimental scope, PVB demonstrates a meaningful and substantial improvement in generating physically realistic trajectories.
>
> - **Q2**: The claimed cross-domain transferability is supported only by protein and protein-ligand tasks; other molecular domains (RNA, materials, polymers) are underexplored.
>
> **A2**: Thank you for the insightful comment. We agree that we have not yet validated transferability to domains such as RNA, materials, or polymers. Importantly, PVB is a fully atomistic, domain-agnostic framework and can in principle be applied to these modalities without architectural changes or extra modules. The primary practical limitation is the scarcity of high-quality dynamical trajectory data in many of these domains, which limits effective training. Addressing this data gap and extending PVB to RNA, materials, and polymers is therefore a clear and important direction for future work.
>
> - **Q3**: I am particularly interested in the experimental results on the large-scale MDCath dataset, as well as the runtime analysis of the proposed method. Could the authors provide more details or quantitative comparisons to illustrate the computational efficiency and scalability of PVB?
>
> **A3**: Thank you very much for your valuable question! First, in terms of the computational efficiency, we have provided the wall-clock time required for one-step inference across different baselines in **Appendix G.4** and **Figure 7**. Our model, PVB, requires on average only about 0.5 seconds per step, achieving a 5-10$\times$ speedup compared to existing baselines.
>
> Second, regarding scaling to larger systems, we are currently conducting experiments on mdCATH [1] to evaluate the performance and scalability of our method. The detailed dataset processing, experimental setup, and results have been updated in **Section 4.1** and **Appendix D**. Briefly, we evaluate PVB on the mdCATH dataset using trajectories simulated at 320 K, following the 5,049/40/20 train/validation/test split from [2]. For evaluation, four of the five independently generated trajectories per test system are used to construct the reference statistics, while the remaining trajectory serves as the MD oracle for direct comparison with the PVB-generated trajectories.
>
> The evaluation results on the 20 test domains from mdCATH have been updated to **Table 2** in the revised manuscript. In addition, Figure 3 presents visualizations for two more mdCATH test systems. For ease of reference, we also include the evaluation results below (**Table A**). Similarly, PVB achieves MD-level physicality in the generated conformations, while also closely capturing the slow dynamical modes and metastable states observed in the MD reference ensembles. Moreover, PVB consistently outperforms UniSim in distributional similarity, physicality, and the proportion of successfully decorrelated trajectories, demonstrating stronger generalization when evaluated on a broader-scope dataset.
>
> Table A. Results on the test set of mdCATH. Values of each metric are shown in mean/std of all 20 test protein domains.
>
> | Models     |      Rg     |    Torus    |     TIC     |     MSM     |    VAL-CA   | RMSE-CONTACT | Decorr-TIC0 |
> |------------|:-----------:|:-----------:|:-----------:|:-----------:|:-----------:|:------------:|:-----------:|
> | MD Oracle  | 0.264/0.085 | 0.182/0.037 | 0.316/0.071 | 0.282/0.069 | 0.942/0.154 |  0.067/0.039 |      -      |
> | UniSim     | 0.566/0.162 | 0.357/0.050 | 0.442/0.105 | 0.369/0.142 | 0.522/0.371 |  0.116/0.041 |     0.30    |
> | PVB (ours) | 0.443/0.143 | 0.347/0.036 | 0.374/0.048 | 0.357/0.136 | 0.963/0.049 |  0.177/0.030 |     0.95    |
>
> (To be continued)

---

> > ### Author Response · Authors · 2025-11-21
> > **Continued**
> >
> > - **Q4**: Is it possible to extend the proposed framework for parallel trajectory generation, rather than sequential sampling? This could further improve scalability, especially for long biomolecular simulations.
> >
> > **A4**: Thank you for your question. As you pointed out, leveraging parallel computing for trajectory generation is an interesting and important direction. Since our method adopts a paradigm that models the transition distribution at coarsened timesteps, the scope for parallelization is currently limited. Nevertheless, we envision the following two possible approaches for parallel processing:
> > 1. Generating multiple independent trajectories from the same initial state in parallel: This leverages the stochastic nature of our SDE-based generative model, where different random seeds can produce distinct transition paths between states. This approach can help rapidly obtain multiple transition pathways and reconstruct the full free energy landscape.
> > 2. Inspired by replica exchange in traditional MD simulations: One could train two models separately using high-temperature and low-temperature trajectory samples, then generate trajectories for the same molecular system in parallel. The high-temperature model can quickly cross energy barriers and access different metastable states. The corresponding high- and low-temperature samples can then be exchanged, allowing the low-temperature model to generate the free energy surface around the newly accessed metastable state, while the high-temperature model restarts exploration of the landscape from different initial states.
> >
> > We also want to emphasize that, even with the current model that allows only serial generation, long-time simulations for larger molecular systems remain feasible. In **Appendix G.1**, we supplement our results with 100,000-step generation (corresponding to 10 μs) for five fast-folding proteins of varying scales and secondary structures. Remarkably, the model achieves high stability without any post-optimization and, in some cases, reconstructs free energy surfaces closely matching those obtained from supercomputer-level MD trajectories. These results demonstrate the stability and reliability of PVB in long-time MD simulations.
> >
> > - **Q5**: Lastly, could the authors elaborate on the role of the latent variable X0? Specifically, in the statement “The latent variable X0 is introduced to avoid the collapse of the conditional probability µ(X1|X0) from degenerating into a Dirac delta measure," it would be helpful to clarify why this latent variable is necessary and what would happen if one directly generated X1 conditioned on X0 without it.
> >
> > **A5**: Thank you very much for your question. Here, I provide a clearer explanation of the role of the latent variable.
> >
> > First, considering the scarcity and limited diversity of MD trajectory data, our goal is to first pretrain the model on a large, diverse, and high-quality set of single-structure data, and then finetune on MD trajectory data to improve the model's cross-domain generalization and generation quality. The main challenge here is that the training objectives in the pretraining and finetuning stages are not consistent.
> >
> > Specifically, during finetuning, the input data are pairs sampled from trajectories $(x_t, x_{t+\tau})$, and the generative model naturally learns to map $x_t$ to $x_{t+\tau}$. In contrast, in the pretraining stage, the input data consist of only one single conformation $x$. If we set the target distribution as that corresponding to $x$, there are two common choices for the prior distribution:
> >
> > 1. If we follow the diffusion model approach and choose a standard Gaussian as the prior, the prior distributions between pretraining (i.e., N(0,1)) and finetuning (i.e., $\mu(x_t)$) will differ substantially. This misalignment prevents the structural knowledge learned during pretraining from being transferable.
> > 2. If the prior distribution is chosen as the same as the distribution of $x$, the generative model is forced to learn a mapping from $x$ to $x$, which degenerates into a trivial case where the model outputs zero residuals.
> >
> > To address this issue, we introduce an intermediate latent state $\tilde{x}$ during pretraining, whose prior is defined as a Gaussian distribution with mean $x$ and standard deviation $\sigma_e$. This effectively decomposes the original mapping into two non-trivial mappings: an encoder that maps $x$ to $\tilde{x}$ and a decoder that maps $\tilde{x}$ back to $x$. In this way, the model in the pretraining stage can follow the same training paradigm as in finetuning on MD trajectory pairs, even when the input is a single structure. This design significantly reduces information loss when transferring from pretraining to finetuning and maximizes the utilization of structural knowledge learned during pretraining. I hope this explanation addresses your question.

---

> > > ### Author Response · Authors · 2025-11-21
> > > **Continued**
> > >
> > > **Reference**
> > >
> > > > [1] Mirarchi, A., Giorgino, T., & De Fabritiis, G. (2024). mdCATH: A large-scale MD dataset for data-driven computational biophysics. Scientific Data, 11(1), 1299.
> > >
> > > > [2] Janson, G., Jussupow, A., & Feig, M. (2025). Deep generative modeling of temperature-dependent structural ensembles of proteins. bioRxiv.

---

> > > ### Comment · Reviewer_BUEn · 2025-11-24
> > > **Response to authors' rebuttal**
> > >
> > > Hi, thank you for the detailed responses, which addressed most of my concerns. I will keep my positive score unchanged.

---

> > > > ### Author Response · Authors · 2025-11-24
> > > >
> > > > Thank you for your thoughtful feedback and positive assessment. We are pleased that our clarifications and additional experiments addressed most of your concerns, and we sincerely appreciate your efforts in helping improve the paper.
> > > >
> > > > Wish you all the best!

---

### Official Review · Reviewer_WXYF · 2025-10-30

**Soundness:** 3
**Presentation:** 2
**Contribution:** 3
**Rating:** 4
**Confidence:** 4

**Summary:**

The paper proposes a method for coarse-grained molecular dynamics simulation, using Brownian bridges. More specifically, they first go to a latent state, and then to the target state. This allows generalization over both single-structure and paired trajectory data, especially in terms of pretraining. Method is evaluated on relevant datasets.

**Strengths:**

- The paper works on a relevant problem.
- Optimal control methods are leveraged for more efficient training.
- The method is evaluated on relevant benchmarks.

**Weaknesses:**

- The explanation of the theory and the notation is quite confusing. I sympathize that this is not trivial, especially with having in a mind a relatively broad target audience from diverse research backgrounds. But considerable effort should be made to improve the writing. I try to make some concrete suggestions below. I'm certainly willing to raise my score if readability is improved!

**Questions:**

- Fig. 1. This can be a very informative figure, but the elements are quite small (the arrows and black dot). Consider indicating on the figure what the meaning is of the three small modes on the left and the big one on the right.
- Why do you use the rmsd, and not the (Gaussian) log likelihood?
- At the start of 3. Method, Z and C are defined, but are not used in the remaining sections. Do you also model these? If so, how exactly?
- Why exactly do you use \mu vs. p? They both indicate probability measures I assume?
- Please define more clearly what \mu, and X are on line 41. In general, please take some time to rethink where you define the different mathematical concepts and objects. Now it's a bit all over the place and does not seem to follow a structured explanation, or logical build up in your story.
- eq. 11, why is there a gradient stop on u when sampling Y_0:1?
- footnote on line 225. Please don't put this in a footnote, it is very confusing! The bridge is from \tilde X to X_1, correct? Why not call it Y from the start, please take some time to think about this, I'm sure there's ways to make this paper much more readable if you decide on certain notations from the start and don't start changing/adding things in the middle of your explanation.
- Prop. 3.2: You are solving an optimal control problem analytically, correct? Is solving the ODE for \tilde a related to solving the HJB equation? Can you please compare your approach to [1], where this is done for the same kind of ELBO, for linear SDEs?

[1] https://arxiv.org/abs/2505.17150


Minor suggestions:
- line 92. RL is not defined.
- line 102: 'applying'
- I would not use (so much) abbreviations in the abstract, it does not improve readability, and RL is not defined. The abstract is quite wordy which makes it also harder to read (e.g. 'inefficiency' -> cost, 'nevertheless', 'remarkable', ..)
- please use larger brackets and ||, e.g. \left[ \right], see eqs. 5, 7, ..
- line 57: 'domain' (singular)
- line 234: 'prove', this sentence is also not clear?
- line 278: Y=Y, very confusing. I suppose one is a stochastic variable and the other one is a realization. You could e.g. use small letters for the realizations.

---

> ### Author Response · Authors · 2025-11-21
>
> We would like to thank the reviewer for the thoughtful reviews and the constructive feedback they provided. In the following, we address the reviewer's questions in detail, aiming to clarify our methodology and further strengthen the presentation of our work.
>
> - **Q1**: Fig. 1. This can be a very informative figure, but the elements are quite small (the arrows and black dot). Consider indicating on the figure what the meaning is of the three small modes on the left and the big one on the right.
>
> **A1**: Thank you very much for your helpful suggestion! **Figure 1** was originally intended as an illustrative diagram, and the different modes shown were randomly generated without specific physical meaning. Following your suggestion, we have explicitly indicated the meaning of each mode in the revised version: the large mode on the right now represents the unfolded state of the protein monomer, and the three smaller modes on the left correspond to three distinct folded states. In addition, as you pointed out, we have adjusted both the arrow thickness and the dot sizes in the figure. The updated figure has already been incorporated into our latest manuscript.
>
> - **Q2**: Why do you use the rmsd, and not the (Gaussian) log likelihood?
>
> **A2**: Thank you for your comment. Since the context of the question is not that clear, we interpret it as referring to the choice of reward function: why we use rmsd instead of a Gaussian log likelihood. Our reasoning is as follows:
>
> Unlike direct supervised learning with the holo state as the training target, our RL finetuning objective aims to increase the probability weight of the holo state in the generated distribution while preserving the model’s ability to produce physically plausible trajectories. Under this setting, the predicted next state may still differ substantially from the holo state. Therefore, assuming that the model prediction error follows a Gaussian distribution with a fixed variance would not be appropriate.
>
> In contrast, RMSD provides a more intuitive and robust measure of the ligand's positional deviation from the true binding pocket. It captures the geometric discrepancy directly of both ligand and protein poses, and serves as a simpler yet effective metric for defining the reward signal in our reinforcement learning setup.
>
> - **Q3**: At the start of 3. Method, Z and C are defined, but are not used in the remaining sections. Do you also model these? If so, how exactly?
>
> **A3**: Thank you for your careful observation. We would like to clarify that Z and C are indeed used as model inputs; their explicit notation was omitted in the later parts of **Section 3** purely for notational brevity, as noted in the footnote on page **5**. The full modeling details are provided in **Appendix C**. In brief, Z is mapped into the latent space through a learnable embedding layer and integrated into the atomic representations, whereas C is encoded into edge features via a lightweight encoder and concatenated with the native edge features of TorchMD-NET. This design ensures that molecular topology is explicitly incorporated into all subsequent message-passing operations.
>
> - **Q4**: Why exactly do you use \mu vs. p? They both indicate probability measures I assume?
>
> **A4**: Thank you for pointing this out. In the original manuscript, our intention was to use $\mu(x_{t+\tau}|x_t)$ to denote the underlying conditional probability measure induced by the MD trajectories, where $\mu_e$ and $\mu_d$ represent the two transition kernels of the underlying Markov process. The model then learns parameterized conditional probability measures $p_e$ and $p_d$ to approximate $\mu_e$ and $\mu_d$, respectively.
>
> We acknowledge that the original notation may have introduced ambiguity. To avoid this, we have revised the notation in the updated manuscript: all symbols $q$ now uniformly denote the target (conditional) probability measures to be learned from data, while all symbols $p$ denote the corresponding parameterized probability measures defined by our model. This update ensures a clearer and more consistent distinction between the underlying distributions and the model-learned ones.
>
> (To be continued)

---

> > ### Author Response · Authors · 2025-11-21
> > **Continued**
> >
> > - **Q5**: Please define more clearly what \mu, and X are on line 41. In general, please take some time to rethink where you define the different mathematical concepts and objects. Now it's a bit all over the place and does not seem to follow a structured explanation, or logical build up in your story.
> >
> > **A5**: Thank you for the helpful suggestion. In the original manuscript, the symbol $\mu$ on line 41 was introduced to denote the Boltzmann probability distribution. Since this definition was not used subsequently, we have removed it in the revised version to avoid unnecessary notation.
> >
> > Regarding your broader comment on the definitions of mathematical concepts and objects, we agree that the notation in the previous submission could be streamlined. In the updated manuscript, we have substantially revised and reorganized the notation to ensure clarity and prevent ambiguities. The key improvements include:
> > 1. Adopting a unified convention that uses $q$ to denote the probability measure associated with the underlying MD process and $p$ to denote the model-parameterized probability measure, thereby avoiding the previous mixture of measures and densities that led to ambiguity.
> > 2. Consistent notation for the three random variables in the Markov chain, now uniformly denoted as $X_0 \to Y_0 \to Y_1$, eliminating the earlier inconsistency when discussing the decoder SDE.
> > 3. Clear distinction between random variables and their realizations by consistently using uppercase and lowercase symbols, respectively.
> > These revisions significantly improve the logical structure and readability of the manuscript. We kindly invite you to review the updated version.
> >
> > - **Q6**: eq. 11, why is there a gradient stop on u when sampling Y_0:1?
> >
> > **A6**: Thank you for your question. The reason for applying a gradient stop during sampling follows directly from the proof in the original Adjoint Matching paper [1], and Proposition 3.2 in our work extends their conclusion to the bridge matching framework. From a practical perspective, performing a gradient stop during trajectory sampling helps prevent gradient accumulation when discretely solving the SDE, which substantially reduces GPU memory consumption.
> >
> > - **Q7**: footnote on line 225. Please don't put this in a footnote, it is very confusing! The bridge is from \tilde X to X_1, correct? Why not call it Y from the start, please take some time to think about this, I'm sure there's ways to make this paper much more readable if you decide on certain notations from the start and don't start changing/adding things in the middle of your explanation.
> >
> > **A7**: Thank you for the insightful comment. We agree that the footnote and the change of notation introduced at that point were confusing in the original submission. In the revised manuscript, we have reorganized the notation to ensure a consistent and coherent presentation throughout.
> >
> > In particular, as part of the broader notation update (see our response to Q5), we now use $X_0, Y_0, Y_1$ as the unified set of random variables for the Markov chain, including in the discussion of the bridge construction for the decoder SDE. This eliminates the need to introduce additional symbols such as $\tilde{X}_0$ or to reinterpret variables mid-exposition. Consequently, the footnote in line 225 has been removed, and the surrounding text has been rewritten such that the bridge is consistently described from the appropriate random variable $Y_0 \to Y_1$ without any change of notation.
> >
> > We believe these revisions significantly improve readability and ensure that the notation remains stable and unambiguous throughout the explanation. We kindly invite you to review the updated version.
> >
> > (To be continued)

---

> > > ### Author Response · Authors · 2025-11-21
> > > **Continued**
> > >
> > > - **Q8**: Prop. 3.2: You are solving an optimal control problem analytically, correct? Is solving the ODE for \tilde a related to solving the HJB equation? Can you please compare your approach to [2], where this is done for the same kind of ELBO, for linear SDEs?
> > >
> > > **A8**: Thank you for your question. Your understanding is correct: Proposition 3.2 indeed concerns solving the SOC problem, where the optimal control $u^*$ is the unique minimizer of Eq. (11). Since its analytical solution is intractable, we parameterize the control $u$ using a neural network.
> > >
> > > **The relation between $\tilde{a}$ and the HJB equation**
> > >
> > > Given $Y_0=y_0$, we first define the cost functional of the SOC problem in eq. (10) as $J(u;y,t)=E_{Y \sim P^u}[\int_t^1(\frac{1}{2}||u(s,y_0,Y_s)||^2)ds-\frac{1}{\beta}r(Y_1) | Y_t=y]$. Solving the corresponding HJB equation is equivalent to finding the optimal control $u^* $ that minimizes this cost functional, i.e., $u^*=\text{argmin}_u J(u;y,t)$.
> > >
> > > Following [1], if we define the "adjoint state" as $a(t,Y,u)=\nabla_{Y_t}(\int_t^1(\frac{1}{2}||u(s,y_0,Y_s)||^2)ds-\frac{1}{\beta}r(Y_1))$, with $Y \sim P^u$, then the conditional expectation satisfies $E_{Y \sim P^u}[a(t,Y,u)|Y_t=y]=J(u;y,t)$, which explicitly connects the adjoint state formulation to the HJB framework.
> > >
> > > By expressing the adjoint state in an ODE form and applying the simplifications described in [1], we ultimately derive the ODE for the so-called "lean adjoint state" $\tilde{a}$ as given in eq. (12). It is guaranteed that the optimal solution $u^*$ to eq. (11) corresponds precisely to the unique minimizer of the HJB equation.
> > >
> > > **Comparison with [2]**
> > >
> > > Thank you for recommending this paper. Our adjoint-matching-based method shares the same optimization problem of the referred method, i.e., solving the SOC problem. The key difference lies in the treatment of the control term: their approach decomposes the control into a linear SDE component and a nonlinear SDE component, where the former admits a closed-form solution derived from the SOC formulation, and the latter is modeled using a neural network. Experiments on financial time-series data demonstrate that such a decomposition accelerates convergence during training. However, its effectiveness has not been validated on higher-dimensional systems.
> > >
> > > In contrast, our adjoint-matching approach does not yield a closed-form solution and still relies on trajectory sampling for optimization. The crucial point, however, is that through a series of equivalent reformulations, our method enables "offline" trajectory sampling. That is, it avoids backpropagating through the discrete SDE solver, which results in a memory-efficient reinforcement learning framework. Moreover, our method has been empirically validated on large-scale protein-ligand systems, demonstrating its scalability to high-dimensional data.
> > >
> > > - **Q9**: Minor Suggestions: line 92. RL is not defined...
> > >
> > > **A9**: Thank you for your careful reading of our manuscript and for your valuable suggestions. All the mentioned issues have been revised in the latest version of the manuscript, which we kindly invite you to review.
> > >
> > >
> > > **Reference**
> > >
> > > > [1] Domingo-Enrich, C., Drozdzal, M., Karrer, B., & Chen, R. T. (2024). Adjoint matching: Fine-tuning flow and diffusion generative models with memoryless stochastic optimal control. arXiv preprint arXiv:2409.08861.
> > >
> > > > [2] Daems, R., Opper, M., Crevecoeur, G., & Birdal, T. (2025). Efficient Training of Neural SDEs Using Stochastic Optimal Control. arXiv preprint arXiv:2505.17150.

---

> > > > ### Comment · Reviewer_WXYF · 2025-11-26
> > > >
> > > > Hi, thanks for all your extra work and detailed responses to all the reviewers. I will raise my score.

---

> > > > > ### Author Response · Authors · 2025-11-26
> > > > >
> > > > > Thank you so much for your thoughtful and constructive review. Your comments on the writing and notation were incredibly helpful and have significantly improved the clarity and quality of our paper. I truly appreciate the time and care you devoted to reassessing our work.
> > > > >
> > > > > Best regards,
> > > > >
> > > > > authors of #10490

---

### Official Review · Reviewer_JWxB · 2025-10-31

**Soundness:** 4
**Presentation:** 4
**Contribution:** 3
**Rating:** 8
**Confidence:** 4

**Summary:**

This paper introduces pretrained variational bridge (PVB). The core contribution is a unified framework that first pretrains an encoder-decoder model on a large and diverse dataset of single, static molecular structures to learn generalizable structural features. This pretrained model is then finetuned on paired molecular dynamics (MD) trajectory data to learn system-specific dynamics. Furthermore, the paper presents RL finetuning procedure using adjoint matching, which efficiently optimizes the model to guide trajectories toward specific target states, such as the holo conformation in protein-ligand docking.

**Strengths:**

- The paper proposes a novel methodology to include pretraining on datasets with static structures but diverse chemical space, and then finetuning on dynamical data. This enables the model to achieve better chemical transferability despite the limited chemical space coverage of the dynamical data.
- The integration of RL with adjoint matching for pose-optimization in docking is a novel application. And the authors have shown the improvement in the ligand pose after the finetuning.
- The model's performance is thoroughly benchmarked across multiple demanding tasks, including protein dynamics, protein-ligand complex dynamics, and holo state exploration. The comparison against several relevant baselines on different datasets demonstrates the effectiveness of the method
- PVB shows outperformance over baselines across most metrics.
- Ablation studies have been performed to show the benefit of pretraining and finetuning procedure

**Weaknesses:**

- While the paper evaluates against other trajectory-based models, it assesses performance on free energy landscapes. While most of the metrics compared in the paper are actually thermodynamic properties, they can be evaluated with i.i.d. (time-agnostic) sampling models. It will be helpful to benchmark against those methods as well.
- In the meanwhile, although the time-dependent model describes dynamics, it's not obvious from the benchmarks and applications shown in the paper why the time-dependence is needed, what is its advantage over i.i.d. sampling model. It will help to justify the motivation if the authors can clarify that (time-dependence makes the model to describe thermodynamics better than i.i.d. sampling model) or show some cases when kinetics/dynamics are of practical interests in applications.
- The rationale behind using two separately finetuned models for the protein (ATLAS) and protein-ligand (MISATO) tasks is not explained. One might expect that a single model finetuned on both could offer better transferability, especially for the protein component of the dynamics.

**Questions:**

- Have the authors checked the physicality of the ligands (and proteins as well)? Not only the bond break or clashes, but also stereochemical errors. Does that get better or worse with RL finetuning?
- The paper claims cross-domain generalization, but it is not specified whether the train/test splits of datasets were performed based on sequence similarity or other metrics to prevent data leakage and rigorously test generalization to unseen protein folds.

---

> ### Author Response · Authors · 2025-11-21
>
> We are grateful to the reviewers for their thorough evaluation and encouraging remarks. We truly appreciate the recognition of our contributions and the insightful questions raised. In the following, we respond point-by-point to each comment and provide additional explanations and experiments where necessary.
>
> - **Q1**: While the paper evaluates against other trajectory-based models, it assesses performance on free energy landscapes. While most of the metrics compared in the paper are actually thermodynamic properties, they can be evaluated with i.i.d. (time-agnostic) sampling models. It will be helpful to benchmark against those methods as well.
>
> **A1**: Thank you for your suggestion! We have added the well-known AlphaFlow [7] as a time-agnostic baseline for the protein trajectory generation task on the ATLAS dataset. It is worth noting, however, that such models do not suffer from autoregressive error accumulation, and therefore the comparison with trajectory generation models in terms of validity metrics may not be entirely fair.
>
> For a fair comparison, following the official repository instructions, we initialized the AlphaFlow model with the pretrained AlphaFold weights and directly finetuned it on our ATLAS training set. The updated experimental results are reported in **Table 1** of the revised manuscript. As a baseline that generates i.i.d. protein ensembles rather than trajectories, AlphaFlow shows advantages over MD ensembles in conformational plausibility and contact-map similarity, but performs worse in distributional similarity, likely because the finetuned model still assigns overly high probability to near-native states, a behavior influenced by the AlphaFold pretraining scheme.
>
> - **Q2**: In the meanwhile, although the time-dependent model describes dynamics, it's not obvious from the benchmarks and applications shown in the paper why the time-dependence is needed, what is its advantage over i.i.d. sampling model. It will help to justify the motivation if the authors can clarify that (time-dependence makes the model to describe thermodynamics better than i.i.d. sampling model) or show some cases when kinetics/dynamics are of practical interests in applications.
>
> **A2**: Thank you very much for your valuable question! Regarding the necessity of introducing time dependence into the model, we believe that the chosen application scenario in our paper, protein-ligand docking, serves as a strong justification.
>
> As demonstrated in prior studies, the protein-ligand blind docking task is notoriously difficult to generalize to unseen complex systems. The main challenge lies in identifying the correct binding pocket from the global protein structure. For time-independent models, this issue remains unavoidable, as they still follow the traditional docking paradigm of large-scale sampling and scoring.
>
> In contrast, our trajectory-based generative model reformulates the problem as a post-optimization of a coarse docking pose. This allows us to leverage the initial conformations predicted by conventional docking software (such as AutoDock Vina) as approximate references for pocket locations, while our RL-based adjoint matching finetuning enables faster and more targeted exploration toward the holo state. Importantly, such RL finetuning is largely ineffective for time-independent models: without conditioning on the current state, increasing the reward weight becomes equivalent to directly supervising with the holo state, which essentially reverts to the conventional regression or generative paradigms used in standard docking tasks.
>
> (To be continued)

---

> > ### Author Response · Authors · 2025-11-21
> > **Continued**
> >
> > In addition to the example provided above, we note that time-dependent generative models offer further advantages over i.i.d. sampling models when capturing thermodynamic or kinetic behaviors of molecular systems. First, a dynamical model produces trajectories that satisfy detailed balance and preserve temporal correlations, enabling the recovery of kinetic quantities such as transition rates, mean first-passage times, or state-to-state fluxes [4]. These quantities cannot be inferred from i.i.d. samples, which lack the sequential structure required to estimate dynamical observables. Second, time-dependent trajectories allow the reconstruction of free-energy profiles using path-based estimators (e.g., reactive flux or transition path sampling), whereas i.i.d. samples only provide equilibrium configurations [5]. Third, many practical applications explicitly require dynamical information: for instance, folding/unfolding pathways of fast-folding proteins, ligand-binding/unbinding processes relevant to drug discovery, and conformational rearrangements in allosteric regulation [6]. Such processes depend critically on transition pathways rather than isolated equilibrium structures. Therefore, a time-dependent generative model is better suited for capturing both equilibrium and non-equilibrium behaviors, while i.i.d. models inherently omit the temporal structure needed for evaluating dynamic properties.
> >
> >
> > - **Q3**: The rationale behind using two separately finetuned models for the protein (ATLAS) and protein-ligand (MISATO) tasks is not explained. One might expect that a single model finetuned on both could offer better transferability, especially for the protein component of the dynamics.
> >
> > **A3**: Thank you for your question! As you correctly pointed out, intuitively, finetuning on both the ATLAS and MISATO datasets could improve the model’s transferability. However, we chose **not** to do so for the following two main reasons:
> >
> > 1. Different simulation environments: The MD simulations in these two datasets are conducted under distinct settings - ATLAS [1] uses GROMACS v2019.4 with the CHARMM36m force field, whereas MISATO [2] employs the gaff2 force field for ligands and AMBER ff14SB67 for proteins. Other simulation parameters (such as temperature) also differ. These discrepancies lead to substantially different distributions of the generated ensembles, thus joint finetuning on both datasets could have adverse effects rather than improving generalization.
> > 2. Fair comparison with baselines: Most baseline methods do not support training on protein-ligand complex trajectories. Therefore, to ensure a fair and consistent comparison, we finetune our model separately on each dataset.
> >
> > - **Q4**: Have the authors checked the physicality of the ligands (and proteins as well)? Not only the bond break or clashes, but also stereochemical errors. Does that get better or worse with RL finetuning?
> >
> > **A4**: Thank you for your valuable suggestions! For the trajectories generated by the model on the 23 test systems of PDBBind v2020, we evaluated the physicality of both protein and ligand conformations and compared the model’s performance before and after reinforcement learning. The detailed experimental setup and results have been updated in **Section G.3** in the revised manuscript.
> >
> > Specifically, for ligands, we evaluated physicality using bond length validity (VAL-BL), bond angle validity (VAL-BA), steric clash free (SCF), and stereochemical integrity (SI). For proteins, we used the same metric as in the monomeric protein trajectory generation task, VAL-CA. The experimental results are provided in **Table A** attached below. From these results, we draw the following three conclusions:
> >
> > Table A. Physicality of the generated protein-ligand complex trajectories on the test set of PDBBind. The metrics are reported in mean/std over 23 test systems.
> >
> > | Models     |    VAL-BL   |    VAL-BA   |    VAL-CA   |     SCF     |      SI     |
> > |------------|:-----------:|:-----------:|:-----------:|:-----------:|:-----------:|
> > | PVB w/o RL | 0.874/0.200 | 0.827/0.212 | 0.615/0.217 | 0.962/0.051 | 0.870/0.344 |
> > | PVB w/ RL  | 0.895/0.204 | 0.852/0.220 | 0.813/0.224 | 0.963/0.070 | 0.870/0.344 |
> >
> > (To be continued)

---

> > > ### Author Response · Authors · 2025-11-21
> > > **Continued**
> > >
> > > 1. First, the model trained on MISATO (i.e., PVB w/o RL) preserves a high degree of physical plausibility in the generated all-atom conformations of both ligands and proteins. This further demonstrates that large-scale pretraining on diverse single structures, followed by light MD finetuning, effectively imparts domain knowledge applicable across molecular families.
> > > 2. Second, after reinforcement learning, the model exhibits slight improvements on ligand-related metrics and a notable enhancement in the validity of generated protein conformations. This indicates that reinforcement learning preserves, and may even enhance, the model’s ability to generate physically plausible trajectories supported by additional data.
> > > 3. Finally, we find that the two models perform identically on the stereochemical integrity metric. Upon inspection, we determined that in 3 of the 23 test systems, the initial ligand conformations from the original database already failed this test. Since the input specifies the molecular topology, the generated trajectories do not alter stereochemistry of the ligand, leading all ligand conformations in the trajectories to fail the check as well. This accounts for the unusual consistency observed for this metric.
> > >
> > > - **Q5**: The paper claims cross-domain generalization, but it is not specified whether the train/test splits of datasets were performed based on sequence similarity or other metrics to prevent data leakage and rigorously test generalization to unseen protein folds.
> > >
> > > **A5**: Thank you for your question! Due to space limitations, we did not provide the full details of dataset construction in the main text, but rather included them in Appendix D in the updated manuscript. As described there, the ATLAS dataset split follows UniSim [3], where the original paper specifies that the training and test sets are divided using a 30% sequence similarity threshold with MMseq2. For the MISATO dataset, we adopt the same train/validation/test split as defined in the original paper, which is based on clustering the amino acid sequences of proteins using BlastP [2]. Considering the computational budget, we further select a subset of 20 complexes from the original test set for evaluation in our study. We hope the above explanation clarifies your concerns.
> > >
> > >
> > > **Reference**
> > >
> > > > [1] Vander Meersche, Y., Cretin, G., Gheeraert, A., Gelly, J. C., & Galochkina, T. (2024). ATLAS: protein flexibility description from atomistic molecular dynamics simulations. Nucleic acids research, 52(D1), D384-D392.
> > >
> > > > [2] Siebenmorgen, T., Menezes, F., Benassou, S., Merdivan, E., Didi, K., Mourão, A. S. D., ... & Popowicz, G. M. (2024). MISATO: machine learning dataset of protein–ligand complexes for structure-based drug discovery. Nature computational science, 4(5), 367-378.
> > >
> > > > [3] Yu, Z., Huang, W., & Liu, Y. UniSim: A Unified Simulator for Time-Coarsened Dynamics of Biomolecules. In Forty-second International Conference on Machine Learning.
> > >
> > > > [4] Tiwary P, Limongelli V, Salvalaglio M, Parrinello M. Kinetics of protein-ligand unbinding: Predicting pathways, rates, and rate-limiting steps. Proc Natl Acad Sci U S A. 2015 Feb 3;112(5):E386-91. doi: 10.1073/pnas.1424461112. Epub 2015 Jan 20. PMID: 25605901; PMCID: PMC4321287.
> > >
> > > > [5] Lazzeri G, Jung H, Bolhuis PG, Covino R. Molecular Free Energies, Rates, and Mechanisms from Data-Efficient Path Sampling Simulations. J Chem Theory Comput. 2023 Dec 26;19(24):9060-9076. doi: 10.1021/acs.jctc.3c00821. Epub 2023 Nov 21. PMID: 37988412; PMCID: PMC10753783.
> > >
> > > > [6] Wolf S, Lickert B, Bray S, Stock G. Multisecond ligand dissociation dynamics from atomistic simulations. Nat Commun. 2020 Jun 10;11(1):2918. doi: 10.1038/s41467-020-16655-1. PMID: 32522984; PMCID: PMC7286908.
> > >
> > > > [7] Jing, B., Berger, B., & Jaakkola, T. AlphaFold Meets Flow Matching for Generating Protein Ensembles. In Forty-first International Conference on Machine Learning.

---

### Official Review · Reviewer_i298 · 2025-11-03

**Soundness:** 3
**Presentation:** 2
**Contribution:** 3
**Rating:** 4
**Confidence:** 4

**Summary:**

The paper proposes a strategy for developing next-step MD emulators by first pretraining on a bridge that maps between distributions of static structures. This allows models trained on vast amounts of rich static data to be easily tuned on dynamics prediction. The authors show that models pretrained on the PDB and PDBBind can be fine-tuned on ATLAS (protein simulations) and MISATO (protein-ligand simulations) to replicate observables. The authors also develop a RL training strategy for the bridge to steer rollouts towards the holo state of a protein-ligand complex.

**Strengths:**

The idea of pretraining a bridge to recapitulate the initial state, and then fine-tuning it to produce the evolved state, is quite interesting. The work also touches upon simulation of protein-ligand simulations, which have been somewhat neglected in the ML for MD literature, despite their significant practical importance.

**Weaknesses:**

**Method**
* The RL formulation of the holo complex finetuning task seems gratuitous. In particular, if the reward is the RMSD to the holo state, why can't the holo state be used in a supervised fine-tuning fashion? If would seem that if the reward is simply the similarity to an explicit, known state, that is the setting of supervised learning, not reinforcement learning.

**Experiments**
* There are missing controls that make the value of the pretraining bridge hard to interpret. What if we pretrain without a bridge, such as AlphaFlow (with templates)? What if we don't pretrain at all, but use the same architecture? (I assume the retrained ITO baseline is using the ITO architecture).
* Although I am willing to judge these as not the focus of the paper, the protein-ligand docking evaluations are extremely sparse - the single baseline is AutoDock Vina, despite vast amounts of recent literature.

**Questions:**

What is the state matrix in Figure 3, right? The caption says "Probability differences between PVB
and MD across the 10 metastable states estimated by MSM." --- this shouldn't be a matrix, then.

---

> ### Author Response · Authors · 2025-11-21
>
> We sincerely thank reviewers for their time, effort, and valuable feedback. We greatly appreciate the positive assessment of our work and the constructive comments that help us further improve the paper. Below, we carefully address the reviewer’s concerns and clarify the points that may have caused confusion.
>
> - **Q1**: The RL formulation of the holo complex finetuning task seems gratuitous. In particular, if the reward is the RMSD to the holo state, why can't the holo state be used in a supervised fine-tuning fashion? If would seem that if the reward is simply the similarity to an explicit, known state, that is the setting of supervised learning, not reinforcement learning.
>
> **A1**: Thank you for your question. We agree that using RMSD directly as a supervised learning objective is a valid approach, commonly used in protein-ligand docking. However, our work focuses on trajectory generation, which requires predicting all intermediate states between the initial and final (holo) structures. This makes our task fundamentally more challenging than docking, which is primarily concerned with the accuracy of the final pose.
>
> Rather than predicting the holo state in a single step, our method treats docking as a post-optimization task. Starting from a rough initial pose (obtained from AutoDock Vina), the PVB model generates a short trajectory, and a confidence model identifies the most probable holo-like state. Because the Euclidean distance between the current and holo states can be very large, a purely supervised objective would heavily distort the model's distribution. In contrast, the RL framework allows for controlled adjustment of the generative distribution: it increases the likelihood of reaching the target state while preserving physically plausible trajectories, enabling the model to efficiently access the desired conformation within a relatively short trajectory.
>
> We hope our response clarifies your question and addresses your concerns.
>
> - **Q2**: There are missing controls that make the value of the pretraining bridge hard to interpret. What if we pretrain without a bridge, such as AlphaFlow (with templates)? What if we don't pretrain at all, but use the same architecture? (I assume the retrained ITO baseline is using the ITO architecture).
>
> **A2**: Thank you for your valuable question! As you pointed out, ablation studies (Table A below) are indeed essential to demonstrate the effectiveness of the pretraining bridge.
>
> Regarding your first question, "What if we pretrain without a bridge", we would like to respond from two perspectives:
>
> First, pretraining on a cross-domain dataset requires both a unified representation across different molecular types (e.g., small molecules, proteins, protein-ligand complexes) and a mechanism to handle the mismatch between the pretraining and finetuning objectives on MD trajectory data, which is precisely the key challenge that our proposed PVB framework aims to address. Therefore, using a framework such as AlphaFlow (specifically designed for proteins) for pretraining may not be applicable, and the pretraining process should still be considered within the context of our PVB framework.
>
> Second, regarding whether it is possible to perform pretraining without the bridge matching component, we tested replacing the PVB decoder's bridge matching entirely with the conditional diffusion model used in ITO, while fully replicating the original pretraining and finetuning scheme. The ablation setup and results are updated in **Appendix G.2**, with the results also provided in **Table A** attached below for reference. The experimental results show that the modified model performs poorly overall and fails to generate physically plausible protein conformations. This suggests, to some extent, the limitations and incompatibility of ITO's generative framework in this setting. We also emphasize that, although bridge matching is not necessarily irreplaceable, it does perform remarkably well under the current experimental setup.
>
> (To be continued)

---

> > ### Author Response · Authors · 2025-11-21
> > **Continued**
> >
> > As for your second question, "What if we don't pretrain at all, but use the same architecture", we have already included relevant results in **Appendix G.2**. Specifically, we compared two ablated variants: (1) **PVB w/o pretraining**, which is trained only on the MD trajectory data, and (2) **PVB w/o finetuning**, which is pretrained on single-conformation data but not finetuned on MD trajectories. It can be clearly observed that without pretraining on single-conformation data, the model’s performance drops significantly across all metrics, particularly in terms of the physical plausibility of generated conformations. This verifies both the effectiveness and the necessity of pretraining on large, diverse and high-quality single-conformation datasets.
> >
> > Table A. Ablation results on the test set of ATLAS. Values of each metric are shown in mean/std of all 14 test proteins. The best result of each metric is shown in **bold**.
> >
> > | Models              |        Rg       |      Torus      |       TIC       |       MSM       |      VAL-CA     |   RMSE-CONTACT  |
> > |---------------------|:---------------:|:---------------:|:---------------:|:---------------:|:---------------:|:---------------:|
> > | PVB                 | **0.478**/0.092 |   0.333/0.029   | **0.366**/0.081 | **0.331**/0.127 | **0.973**/0.021 |   0.146/0.021   |
> > | PVB w/o finetuning  |   0.561/0.123   | **0.310**/0.018 |   0.463/0.094   |   0.407/0.155   |   0.968/0.039   | **0.097**/0.014 |
> > | PVB w/o pretraining |   0.710/0.063   |   0.481/0.011   |   0.408/0.072   |   0.581/0.127   |   0.022/0.007   |   0.223/0.033   |
> > | PVB-ITO             |   0.676/0.082   |   0.577/0.014   |   0.487/0.072   |   0.460/0.158   |   0.000/0.000   |   0.269/0.042   |
> >
> > - **Q3**: Although I am willing to judge these as not the focus of the paper, the protein-ligand docking evaluations are extremely sparse - the single baseline is AutoDock Vina, despite vast amounts of recent literature.
> >
> > **A3**: Thank you very much for your question. We believe there may be a slight misunderstanding regarding the scope of our evaluation, and we would like to clarify it in detail.
> >
> > As already responsed to Q1, our setting differs fundamentally from standard protein-ligand docking benchmarks. Because MD simulations of protein-ligand complexes, including those generated by PVB, exhibit strong locality over short timescales, the generated ligand conformations fluctuate only within a narrow neighborhood of the initial state. As a consequence, it is essentially infeasible to recover a correct holo conformation starting from a random initialization. For this reason, our goal is **not** to solve the full docking problem from scratch. Instead, we explicitly formulate a **post-optimization** task, where PVB refines an already reasonable, but imperfect, docking pose.
> >
> > To construct this setting, we use AutoDock Vina to generate the initial poses, which serve as "coarse but near-native" starting conformations. From each such initialization, PVB produces a short trajectory (100 frames), and a pretrained confidence model selects the most probable holo-like structure from the trajectory. As shown in **Table 4**, the RL-finetuned PVB consistently and substantially improves the ligand poses over the Vina initialization, demonstrating that PVB functions as an effective docking-pose post-optimizer.
> >
> > Under this formulation, results from other docking software are **not directly comparable**, because:
> > 1. Other docking software would generate an alternative initialization based on their own scoring functions, and therefore their outputs cannot be fairly compared to the poses obtained by refining the AutoDock Vina initialization using our PVB model.
> > 2. The objective of our experiment is specifically to assess whether RL-finetuning enables PVB to reliably and consistently enhance the conformations produced by a given docking tool, rather than to compare absolute docking accuracy across diverse docking algorithms.
> >
> > We hope this clarification makes the evaluation protocol and the rationale behind the chosen baseline clear.
> >
> > - **Q4**: What is the state matrix in Figure 3, right? The caption says "Probability differences between PVB and MD across the 10 metastable states estimated by MSM." --- this shouldn't be a matrix, then.
> >
> > **A4**: Thank you very much for your careful observation! You are absolutely right: the "Probability differences between PVB and MD across the 10 metastable states estimated by MSM" is indeed a vector of length 10. In Figure 3, we present it in a 10×10 matrix format purely for visual aesthetics, where the diagonal entries represent the probability differences for each corresponding metastable state. The off-diagonal elements do not carry any specific meaning and are therefore shown as blank.

---

### Author Response · Authors · 2025-11-21
**General Response**

We sincerely thank all reviewers for their time, constructive feedback, and thoughtful evaluation of our work. We greatly appreciate the reviewers' insightful comments, which have helped us substantially improve the clarity, rigor, and completeness of the manuscript. Below, we provide a consolidated response addressing the major concerns shared by reviewers and summarizing the key revisions made in the updated version of the paper. All revised portions of the manuscript have been **highlighted in red** to facilitate the reviewers' assessment.
- In response to **reviewer WXYF's** concerns about notational problems that caused confusion, we have made substantial revisions throughout the manuscript. The main changes include: (i) adopting a unified convention that uses $q$ to denote the probability measure associated with the underlying MD process and $p$ to denote the model-parameterized probability measure, thereby avoiding the previous mixture of measures and densities that led to ambiguity; (ii) standardizing the notation for the three random variables in the Markov chain as $X_0 \to Y_0 \to Y_1$, thereby removing earlier inconsistencies in the discussion of the decoder SDE; and (iii) clearly distinguishing random variables from their realizations by consistently using uppercase and lowercase symbols, respectively.
- Regarding **reviewer i298's** concerns about the necessity of our pretrained bridge, we would like to clarify that **Appendix G.2** already includes two ablation studies: (i) training the model from scratch directly on MD data without any pretraining on single-structure data, and (ii) generating trajectories using the pretrained model without finetuning on MD data. These experiments respectively demonstrate that pretraining provides a substantial and consistent improvement in the model's generative capabilities, and that finetuning is essential for enhancing the distributional similarity between the generated ensemble and the target MD ensemble. In addition, to address the question of whether using bridge matching in the decoder is reasonable and necessary, we further added an ablation where we replace our bridge-matching decoder with the conditional diffusion model employed in ITO, and run the full pretraining and finetuning pipeline. The experimental setup and results have been updated in **Appendix G.2** as well.
- Regarding **reviewer JWxB's** suggestion to include a time-agnostic baseline, we have added AlphaFlow [1], a well-known model capable of generating i.i.d. protein ensembles. To ensure a fair comparison with our approach, we finetuned AlphaFlow on our ATLAS training set using the pretrained AlphaFold weights provided in its official repository. **Section 4.1** now includes the full experimental setup and the corresponding results on ATLAS.
- Regarding **reviewer JWxB's** concerns about the physicality of the generated protein and ligand conformations, we have conducted additional evaluations following the valuable suggestion. For the trajectories generated by our model on the 23 test systems from PDBBind v2020, we chose appropriate metrics and assessed the physical plausibility of both protein and ligand structures and compared the model's behavior before and after reinforcement learning. The detailed experimental setup and corresponding results have been added to **Section G.3** of the revised manuscript.
- Regarding **reviewer BUEn's** concern that the scope of our chosen datasets may be insufficient, we have expanded our evaluation by conducting additional protein trajectory generation experiments on the mdCATH dataset [2]. The detailed experimental setup and results have been incorporated into **Section 4.1** and **Appendix D** of the revised manuscript.

In summary, we have carefully addressed major concerns raised by the reviewers, including notational clarity, the necessity and design of our pretrained bridge, inclusion of additional baselines, assessment of physicality, and expansion to larger datasets. The corresponding revisions and additional experiments have been clearly highlighted in the manuscript. Responses to other specific comments are provided in the individual replies to each reviewer. We believe these changes substantially strengthen the clarity, rigor, and impact of our work.

**Reference**

> [1] Jing, B., Berger, B., & Jaakkola, T. AlphaFold Meets Flow Matching for Generating Protein Ensembles. In Forty-first International Conference on Machine Learning.

> [2] Mirarchi, A., Giorgino, T., & De Fabritiis, G. (2024). mdCATH: A large-scale MD dataset for data-driven computational biophysics. Scientific Data, 11(1), 1299.

---

### Author Response · Authors · 2025-11-29
**Response to Area Chair**

Dear Area Chairs,

Thank you very much for your time and effort in reviewing our submission. For your convenience, we provide below a concise overview of the major concerns raised by each reviewer and how we have addressed them.

**Reviewer i298**
> Choice of reinforcement learning instead of supervised learning.

In **Section 3.3**, we introduce a reinforcement learning (RL) framework with an explicit reward function to accelerate exploration of the holo state. The reviewer asked why RL is necessary given that the holo state is known, and why supervised learning would not suffice.

In our response, we clarified that protein-ligand docking, especially the precise localization of the ligand, is extremely challenging in the literature, and supervised learning largely falls back to existing paradigms that struggle with this problem. We instead reformulate the task as post-optimization of a coarse docking pose (e.g., from AutoDock-Vina), using short MD rollouts initialized from such poses to iteratively refine the structure. This requires a model that can both explore toward the target binding conformation and generate physically plausible trajectories. Under these requirements, RL provides a highly appropriate paradigm.

> Insufficient ablation study for the pretrained bridge.

The reviewer was concerned that the value of our pretrained bridge is not sufficiently validated. We first pointed out that the original manuscript already includes two ablations in **Appendix G.2**: (i) training on MD trajectories from scratch without pretraining, and (ii) using the pretrained model to generate trajectories without finetuning. These experiments demonstrate the necessity and effectiveness of pretraining, as well as the benefits of finetuning on MD data to better fit the empirical conformational distribution.

Additionally, to address the reviewer's suggestion, we expanded **Appendix G.2** with a new ablation that fully replaces our decoder's bridge matching component with the conditional diffusion model used in ITO. The results show that our current architecture performs better, supporting the rationality of our model choice.

**Reviewer JWxB**

> Request for time-agnostic sampling baselines.

In response, we added the well-known AlphaFlow as an additional baseline. The experiments and results have been incorporated into **Section 4.1** and **Table 1**. It is worth noting that time-agnostic sampling models are not affected by autoregressive error accumulation, and therefore some of their comparisons with trajectory generation models are not entirely fair. We have clarified this both in our responses to the reviewers and in the revised manuscript.

> Request to evaluate the physicality of generated protein-ligand structures.

We added physicality-related metrics and further evaluated the trajectories we previously generated on MISATO, comparing these metrics before and after reinforcement learning. The updated results are provided in **Appendix G.3**.

**Reviewer WXYF**

> Issues with probability-related notation.

The reviewer noted inconsistencies and mixing of concepts related to probability densities, measures, and random variables, which hindered readability. We thoroughly revised the mathematical notation in the updated manuscript and provided a detailed response.

Importantly, during the discussion phase, the reviewer expressed satisfaction with our revisions and increased their score.

**Reviewer BUEn**

> Scope of datasets and suggestion to include mdCATH.

Following the reviewer's recommendation, we added experiments on mdCATH, using UniSim as the baseline. The setup, results, and visualizations have been incorporated into **Section 4.1**, **Table 2**, and **Figure 3**.

> Request for computational efficiency of model inference.

We clarified that **Appendix G.4** of the original manuscript already reports inference efficiency comparisons between our model and the baselines, showing a **5-10**$\times$ speedup over the second-best method, MDGEN.

During the discussion phase, the reviewer expressed satisfaction with our response and maintained their positive score.

For additional clarification, detailed explanations, and point-by-point responses, please refer to our general response and individual responses to each reviewer. Thank you again for your time and consideration!

Best regards,

authors of #10490

---

### Meta-Review · Area_Chair_2LZk · 2026-01-08

**Summary:**

The main concerns are as follows.

1) The selection of reinforcement learning instead of supervised learning. If the reward is simply the similarity to an explicit, known state, that is the setting of supervised learning, not reinforcement learning.

2) Needs to interpret the value of the pretraining bridge.

3) The experiments needs improvement. More baselines should be compared. And more experiments on larger datasets and other molecular domains should be conducted to make the results more convincing.

4) The reasons that the time-dependence is needed should be strengthened and clarified. And the reasons that using two separately finetuned models for the protein (ATLAS) and protein-ligand (MISATO) tasks is not explained.

5) The writing should be improved. Many details are not very clear.

**Reviewer Concerns:**

For the selection of reinforcement learning and supervised learning, the authors should clarify the paper and avoid confused understanding for the readers.

**Reviewer Scores:**

Two reviewers gave 8 score.
One reviewer gave 4 score, but have no further response after author rebuttals.
On reviewer gave 4 score originally, but agree to raise the score. I read the reviews and rebuttals. I think the final score should be 6.

---

### Decision · Program_Chairs · 2026-01-26

Accept (Poster)